# Post-replicative pairing of sister *ter* regions in *Escherichia coli* involves multiple activities of MatP

Estelle Crozat [1,2✉], Catherine Tardin[3], Maya Salhi [1], Philippe Rousseau [1], Armand Lablaine[1], Tommaso Bertoni [2], David Holcman[4], Bianca Sclavi [5], Pietro Cicuta [2] & François Cornet [1]

The *ter* region of the bacterial chromosome, where replication terminates, is the last to be segregated before cell division in *Escherichia coli*. Delayed segregation is controlled by the MatP protein, which binds to specific sites (*matS*) within ter, and interacts with other proteins such as ZapB. Here, we investigate the role of MatP by combining short-time mobility analyses of the *ter* locus with biochemical approaches. We find that *ter* mobility is similar to that of a non *ter* locus, except when sister *ter* loci are paired after replication. This effect depends on MatP, the persistence of catenanes, and ZapB. We characterise MatP/DNA complexes and conclude that MatP binds DNA as a tetramer, but bridging *matS* sites in a DNA-rich environment remains infrequent. We propose that tetramerisation of MatP links *matS* sites with ZapB and/or with non-specific DNA to promote optimal pairing of sister *ter* regions until cell division.

[1] Centre de Biologie Intégrative de Toulouse (CBI Toulouse), Laboratoire de Microbiologie et de Génétique Moléculaires (LMGM), Université de Toulouse, CNRS, UPS, Toulouse, France. [2] Cavendish Laboratory, University of Cambridge, Cambridge CB3 0HE, UK. [3] Institut de Pharmacologie et de Biologie Structurale (IPBS), Université de Toulouse, CNRS, UPS, Toulouse, France. [4] Ecole Normale Supérieure, Applied Math and Computational Biology, IBENS, 46 rue d'Ulm, 75005 Paris, France. [5] Laboratory of Computational and Quantitative Biology (LCQB), UMR 7238 CNRS, Sorbonne Université, 4 Place Jussieu, 75005 Paris, France. ✉email: eb652@cam.ac.uk

Bacterial nucleoid structuration is due to a variety of processes including DNA supercoiling, proteins and complexes working on the DNA (e.g., RNA polymerases), nucleoid-associated proteins and condensins (SMCs)[1]. These act together to shape the chromosomes in a dynamic structure while keeping DNA accessible to polymerases and repair proteins. Replication and segregation re-organise nucleoids on a large scale[2,3]. Whilst some details may vary, common to all bacteria is the bidirectional replication of the chromosome, starting at a unique origin and finishing in the opposite terminus region, which is the last to be segregated before cell division.

Macrodomains are large regions with specific cellular positioning that contain either the origin or terminus of replication[4–8]. In *Escherichia coli* (*E. coli*), the Ter macrodomain (*ter*) is very distinctive (Fig. 1a), spreading along 800 kb encompassing the replication terminus region[7]. It contains 23 *matS* sites that are recognised by the MatP protein[9]. MatP is known to play a key role in *ter* positioning and setting its segregation pattern: it keeps the sister *ter* regions paired near the divisome, at mid-cell, allowing their processing by the DNA-translocase FtsK[10]. Since FtsK activity is oriented by KOPS DNA motifs, segregation ends at the *dif* site, where final unlinking of sister chromosomes occurs[10,11].

How MatP achieves its functions is currently unclear. MatP forms dimers in the absence of DNA, and tetramers upon binding to *matS*-containing DNA[12]. This was proposed to pair *matS* sites, forming large chromosome loops, though they were not detected in contact maps of the chromosome[13]. MatP was also shown to interact with the divisome-associated ZapB protein[14] and the condensin MukB[15]. A truncated variant of MatP (deletion of the last 20 residues), MatPΔ20, was reported unable to form tetramers[12] nor to interact with ZapB[14], yet retaining interaction with MukB[15]. MukBEF was reported to promote long-range interactions between chromosome loci, probably by forming loops, in a MatP-dependent manner[13,16]. Consistent with this view, MatP and MatPΔ20 lower long-range interactions and/or promote short-range interactions inside *ter*[13] while excluding MukB from *ter*[15]. Since MukB interacts with TopoIV, its exclusion by MatP probably delays decatenation of sister *ter*, thus coupling their segregation with cell division[15]. Interaction of MatP with ZapB has been proposed to induce a positive control of divisome assembly around *ter*, (the Ter-linkage)[17,18]. Mutation of *zapB*, as well as *matPΔ20*, alters the mid-cell positioning of MatP-bound sister *ter* and shortens the co-localisation times of *ter* loci[12,14].

Foci movements have also been recorded at different time scales, revealing important differences between chromosome regions. At long time scales, loci tracking captures their segregation dynamics[19–24]. Loci of *ter* localise accurately at mid-cell[19], then separate when early divisome components have formed a complex at mid-cell. At short-time intervals, they sub-diffuse[21,25–27], reflecting constraints imposed by their environment[9,13,27]. A previous study showed that the mobility of loci varied depending on chromosomal localisation[27], the *ter* loci being less mobile when located at mid-cell.

In this report, we investigate the role of MatP in constraining the mobility of a *ter* locus. Surprisingly, low-fluorescence foci of the *ter* locus are as mobile as those of an oriC-proximal locus, showing that the higher constraint of the *ter* locus is not an intrinsic property but depends on context. We further show that highly intense and poorly mobile foci form most often at the *ter* locus and depend on the presence of MatP, suggesting they contain pairs of unsegregated sister loci. This effect depends on MatP, its 20 C-terminal residues and ZapB to different levels. We characterise MatP/DNA complexes and conclude that while MatP binds DNA as a tetramer, it rarely forms specific DNA loops by bridging *matS* sites in a DNA-rich environment, suggesting that the tetramers play a different role.

## Results

**Monitoring chromosome loci mobility in vivo.** To monitor the mobility of chromosome loci, we used strains carrying a *parS* site inserted on the chromosome and producing cognate ParB-GFP proteins[28] (Fig. 1a). We recorded the position of foci every 0.5 s during 20 s (Fig. 1b), then extracted the mean squared displacements (MSD) from these trajectories. An example of 30 MSD for Ter4 is shown in Fig. 1c.

We first used the P1 *parS* site and ParB protein to tag loci in the *ori* and *ter* regions, and we reproduced published results for Ori2 and Ter3[27] (Fig. 1e). However, this P1-derived system has been reported to increase post-replicative cohesion of tagged loci[10,28,29]. We thus tested another set of strains, with loci tagged with *parS*-pMT1 and producing a ParB-pMT1-GFP[28]. The comparison revealed important differences: (1) The number of cells with a single focus decreased and those with two foci increased when using the pMT1-derived system (Fig. 1d); (2) The MSDs obtained with the pMT1 system were larger than with the P1 system (Fig. 1e); (3) The difference in MSDs between the *ori* and *ter* loci were largely reduced when using the pMT1-derived system (Fig. 1e); (4) For a same intensity, a remarkable drop of mobility was observed for Ter loci labelled with *parS*-P1[27]. The P1-derived system thus not only delays *ter* segregation, but creates aggregates of proteins that result in brighter foci (Supplementary Fig. 1) with very low mobility, biasing the results obtained. We chose to proceed with the pMT1-derived system in this work.

**The mobility of a *ter* locus depends on foci intensity.** We next analysed the fate of foci formed at Ori2 and Ter4. Foci populations were binned into categories depending on their number per cell and their localisation, for which we defined two categories: (M) mid-cell (0–1/6th cell length from the cell centre) and (R) rest of the cell (1/6th–1/2). Consistent with previous reports[29–31], segregated foci of Ori2 preferentially localised at the quarter positions, and near mid-cell for single foci. Also consistent with previous reports, Ter4 foci were preferentially located at mid-cell (Supplementary Fig. 2), unsegregated foci being closer from mid-cell than segregated ones.

Foci intensity varied between loci. Ori2 foci followed a sharp distribution centred around 500 AU (Fig. 2a). In contrast, Ter4 foci followed a wider distribution with more high-intensity foci. For moderately intense foci (below 1000 AU), we observed a monotonous decrease of mobility when intensity increased (Fig. 1f). Strikingly, Ori2 and Ter4 foci had equivalent MSDs at corresponding intensities. The low mobility of *ter* loci is thus not an intrinsic property of *ter* but depends on the intensity of the foci (Fig. 1f). At intensities above 1000 AU, foci mobility no longer varied in a monotonous way with increasing intensity and was clearly different between Ori2 and Ter4. From this observation, we defined two categories of foci: foci of low intensity, hereafter called FL, below 1000 AU, and foci of high intensity, hereafter called FH, above 1000 AU. FH were rare at Ori2 (2%) but rather frequent at Ter4 (30%) (Fig. 2b). Double FH of Ter4 were rare and tended to position like double FL.

Calibrating the number of GFP molecules in foci (from the increase of variance in intensity along time) gave an estimated mean of 33 GFP molecules for Ter4 and Ori2 FL, whereas the mean for Ter4 FH was 70 GFP (Supplementary Fig. 3 and "Methods").

**FL and FH show different dynamics.** We analysed the trajectory of foci using four parameters characterising the nature of movement ("Methods"[32]): (1) The anomalous exponent ($\alpha$) is computed from the MSD behaviour for small increments. It

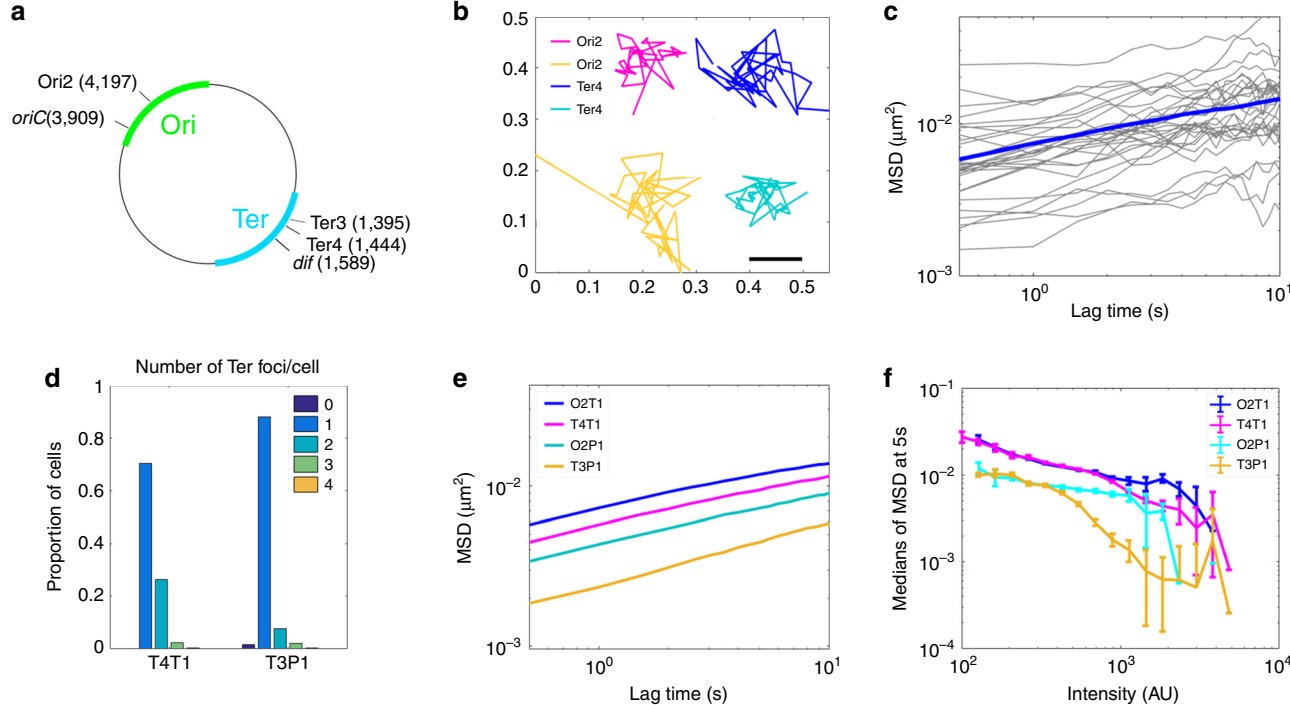

**Fig. 1 The type of visualisation system affects the mobility of chromosomal loci. a** Scheme of *E. coli* chromosome showing two macrodomains, the origin of replication *oriC* and the site-specific recombination site *dif*, plus the loci where *parS* sites have been inserted, called Ori2, Ter3 and Ter4. Positions are indicated in Mbp. **b** Examples of tracks for Ori2 and Ter4 loci; 2 fps for 20 s, parB-pMT1 system. **c** Example of 30 MSD from Ter4 (parB-pMT1) locus as a function of lagtime (grey lines), with the median for all MSD of this experiment ($n = 1600$) shown in blue. **d** Number of Ter foci per cell (dark blue, cells with no focus; orange, cells with four foci), with ParB-P1 (T3P1) or parB-pMT1 (T4T1). *Y* axis represents the proportion of cells with that number of foci compared to the total number of cells. **e** Medians of MSDs as a function of lagtime for Ori2-ParB-P1 (O2P1), Ori2-ParB-pMT1 (O2T1), Ter3-ParB-P1 (T3P1) and Ter4-ParB-pMT1 (T4T1). **f** Medians of MSD of ParB-P1 and ParB-pMT1 foci at 5 s as a function of intensity of the foci. Error bars show the SEM. For each intensity bin, $n = 0, 21, 109, 542, 1512, 3129, 3915, 2876, 1542, 509, 138, 37, 13, 3, 3, 1, 0, 0, 0, 0$ foci for O2T1; $n = 2, 9, 30, 115, 360, 878, 1399, 1769, 1851, 1635, 1173, 602, 249, 57, 14, 6, 1, 0, 0, 0$ for T4T1; $n = 0, 4, 26, 125, 363, 665, 830, 638, 372, 152, 36, 13, 4, 1, 0, 1, 0, 0, 0, 0$ for O2P1 and $n = 0, 3, 20, 84, 231, 395, 532, 514, 438, 317, 216, 88, 71, 26, 10, 2, 1, 0, 0, 0$ for T3P1. Source data are provided as a Source Data file.

indicates the nature of the locus motion. $\alpha = 1$ describes normal diffusion, $\alpha < 1$ is sub-diffusive (constrained) and $\alpha > 1$ is super-diffusive (directed) movement. (2) The length of confinement ($L_c$) is the standard deviation (SD) of the locus position with respect to its mean averaged over time. This estimates the apparent radius of the volume explored by a finite trajectory. (3) The diffusion coefficient ($D_c$) reflects the second-order statistical properties of a trajectory, and accounts for local crowding that may vary along the trajectory. (4) The effective spring coefficient ($K_c$) represents an external force acting on a locus. It is modelled as a spring force applied on a single monomer belonging to a polymer. This force affects the entire polymer motion and can be recovered from the first-order moment statistics of single locus trajectories.

Considering only FL (Fig. 2d), the value of each parameter was poorly dependent of their position, suggesting that loci properties do not change significantly during the cell cycle. Ori2 foci had a low value of $\alpha$ (0.16), whereas Ter4 showed a slightly higher $\alpha$ (0.2, $p = 3.8 \times 10^{-13}$), suggesting that the local condensation is slightly higher in *ori* than in *ter*, according to the RCL model[32–34]. The length of confinement for both loci was small (0.084 and 0.081 μm for Ori2 and Ter4, respectively), revealing loci are confined in small regions, which size depends slightly on chromosomal or cellular location. The diffusion coefficient was higher for Ori2 than for Ter4 ($3.5 \times 10^3$ and $2.9 \times 10^3$ μm² s⁻¹, respectively, $p = 2.7 \times 10^{-22}$), showing that despite a greater condensation, Ori2 is freer to diffuse than Ter4. Finally, the spring coefficient revealed equivalent forces

tethering both Ori2 and Ter4 (331 and 319 $k_B$ T μm⁻²). In cells harbouring two foci, 2FL showed the same behaviour as 1FL (Supplementary Fig. 4a).

Ter4 FH behaved differently than FL (Fig. 2d). Ter4 FH $\alpha$ was close to the FL value (0.22 for FH and 0.2 for FL, $p = 7 \times 10^{-5}$) and remained the same for both cell positions ($p = 0.5$), suggesting a monotonous condensation of the *ter* region. However, $L_c$ was lower than for FL ($p = 10^{-125}$) and lower at mid-cell than in the rest of the cell ($p = 8 \times 10^{-4}$). This suggests Ter4 FH are more confined than FL and more confined when at mid-cell. $D_c$ showed the same trends as $L_c$, showing that Ter4 FH are diffusing less than FL ($p = 10^{-200}$) and less when in the mid-cell area ($p = 5 \times 10^{-5}$). These changes are consistent with a doubling in the force applied to Ter4 FH when at mid-cell (620 $k_B$ T μm⁻² for FH and 331 $k_B$ T μm⁻² for FL, $p = 10^{-130}$).

Ter4 FH are thus, compared to FL: brighter, submitted to twice the force, more confined, less diffusive and preferentially located at mid-cell. A straightforward hypothesis is that they mostly contain sister loci held together in post-replicative cohesion. This is consistent with the extended cohesion period reported for sister *ter* regions[21,29,30], even when the pMT1 system was used[10]. This also explains the higher percentage of FH for Ter4 compared to Ori2. It follows that most FL contain single copies of loci. Assuming this hypothesis, a strong constrain is applied to *ter* loci only when they are in post-replicative cohesion. The rest of the time, their dynamics is much more similar to the one of other chromosome regions than previously thought.

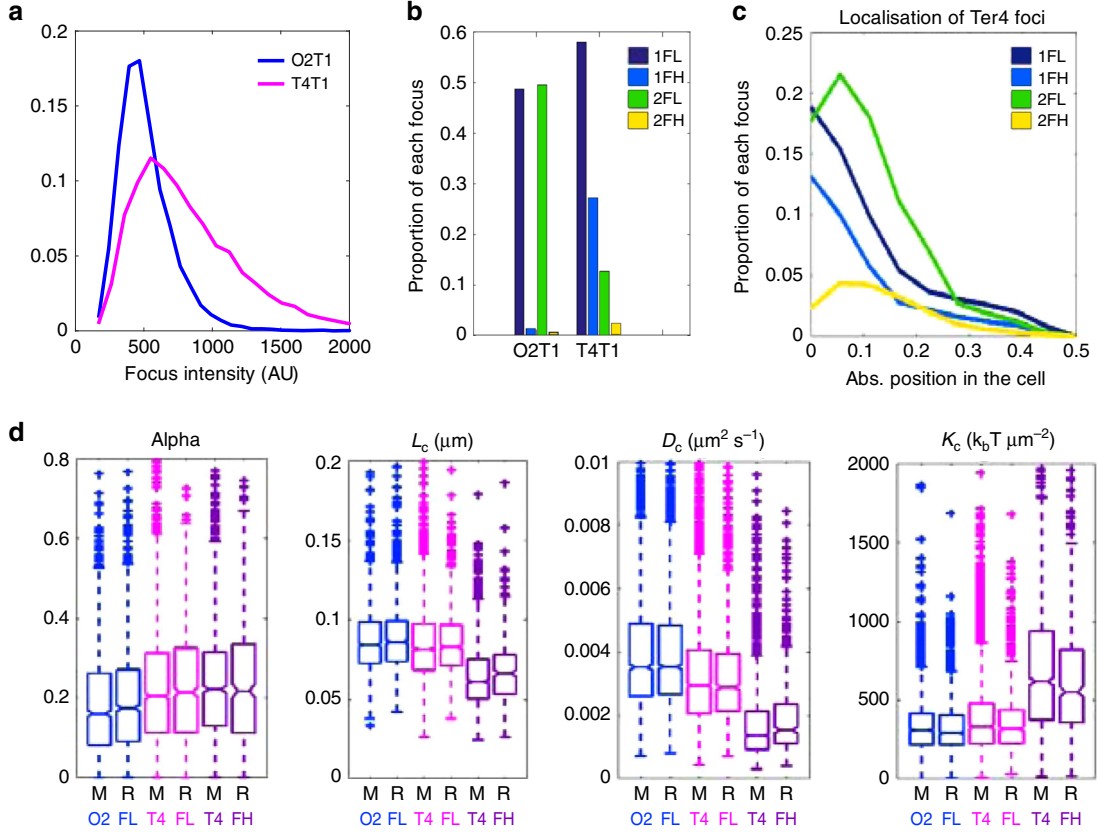

**Fig. 2 The high intensity Ter4 foci display a slower dynamic. a** Distribution of intensity of Ori2 and Ter4 *parS*-pMT1 foci. **b** Proportion of each type of focus in Ori2 and Ter4 strains. 1FL: single, low intensity focus in the cell; 1FH: single, high intensity focus in the cell; 2FL: low intensity focus detected in a cell with two foci; 2FH: high intensity focus detected in a cell with two foci. The numbers of foci of each type are indicated in Supplementary Data 1. **c** Proportion of Ter4 foci as a function of their position in the cell. Zero is mid-cell, while 0.5 is a cell pole. **d** The four parameters $\alpha$, $L_c$, $D_c$ and $K_c$ were calculated for Ori2 and Ter4 foci, which were divided first in low (FL) or high intensity (FH) foci, then further in foci localised at mid-cell (M, 0–0.16) or away from mid-cell (R, 0.17–0.5). Ori2 1FL (blue, $n = 1411$ for M, $n = 1889$ for R), Ter4 1FL (pink, $n = 2744$ for M, $n = 1064$ for R) and Ter4 1FH (purple, $n = 1374$ for M, $n = 412$ for R) are shown. The number of 1FH for Ori2 is too low ($n = 88$) to give significant results and is therefore not plotted here. Box plots show the median of the distribution, the 25th and 75th percentiles of the samples (respectively the bottom and top of the box), the lowest and top values within a range of 1.5 times the interquartile range (dotted lines), and outliers of these as crosses. The notches display the variability of the median; non-overlapping notches between samples indicate different value of the medians at a 5% significance level. Source data are provided as a Source Data file.

**MatP is required for FH formation and maintenance in *ter*.** We next deleted *matP* from our strains and observed the fate of Ter4 foci. The strongest effect was on FH. Cells with one FH decreased drastically (from 27 to 8%; Fig. 3a), whereas cells with one FL decreased moderately (from 58 to 48%). Consistently, cells with two foci increased and mostly contained FL. This increase in two-foci cells is consistent with MatP acting to keep sister *ter* regions together after replication[9]. The large decrease in 1FH cells thus confirms that FH contain pairs of unsegregated loci. In the Δ(*matP*) strain, the remaining FH were more mobile and less confined than in the wild type. They showed increased $L_c$ and $D_c$ and decreased $K_c$ (Fig. 3b, $p = 10^{-8}$, $p = 1.9 \times 10^{-12}$, $p = 7.6 \times 10^{-11}$, respectively). These values are close to the ones obtained for Ori2 FH (Supplementary Data 2, $p = 0.75$, $p = 4 \times 10^{-3}$, $p = 0.56$ for $L_c$, $D_c$ and $K_c$). In addition, FH at mid-cell were not different anymore from the ones located in the rest of the cell (Fig. 3b, $p = 0.6$, $p = 0.4$, $p = 0.5$ for $L_c$, $D_c$ and $K_c$). We conclude that MatP is required both for the high number of FH Ter4 and for their specific constrain at mid-cell. A slight but significant difference in mobility and confinement between FH and FL in the Δ(*matP*) strain persisted (Fig. 3; $p = 4 \times 10^{-17}$, $p = 2 \times 10^{-33}$ and $p = 4 \times 10^{-14}$ for $L_c$, $D_c$ and $K_c$). Remaining FH may be rare single loci with high fluorescence intensity, which should reduce their mobility[35]. However, a fraction

of remaining FH certainly contains foci present on chromosome dimers, which may pair during FtsK/Xer resolving process, since this is MatP-independent[9]. Interestingly, deletion of *matP* had only a slight effect on α of Ter4 FH ($p = 0.02$) and none on FL ($p = 0.06$), suggesting that MatP only marginally influences the local condensation of *ter* DNA.

Deleting *matP* also had a small but significant effect on Ter4 FL dynamics (Fig. 3c). They were less confined and more mobile in the Δ(*matP*) strain (higher $L_c$ ($p = 3 \times 10^{-7}$) and $D_c$ ($p = 6 \times 10^{-7}$), lower $K_c$ ($p = 7 \times 10^{-10}$)). This appears as a general effect, as equivalent variations were observed with Ori2 (Supplementary Fig. 5b).

**Different activities of MatP are required for FH formation.** We then used two mutants: a deletion of *zapB*, coding for the divisome-associated protein interacting with MatP[14,36], and *matP*Δ20, coding for a version of MatP deleted of the last 20 C-terminal residues, defective for both interaction with ZapB and tetramerisation[12]. In these two mutants, MatP still binds *matS* (ref. [12], and below) and interacts with MukB[15].

Both Δ(*zapB*) and *matP*Δ20 mutants showed a phenotype intermediate between the wild-type and Δ(*matP*) strains considering

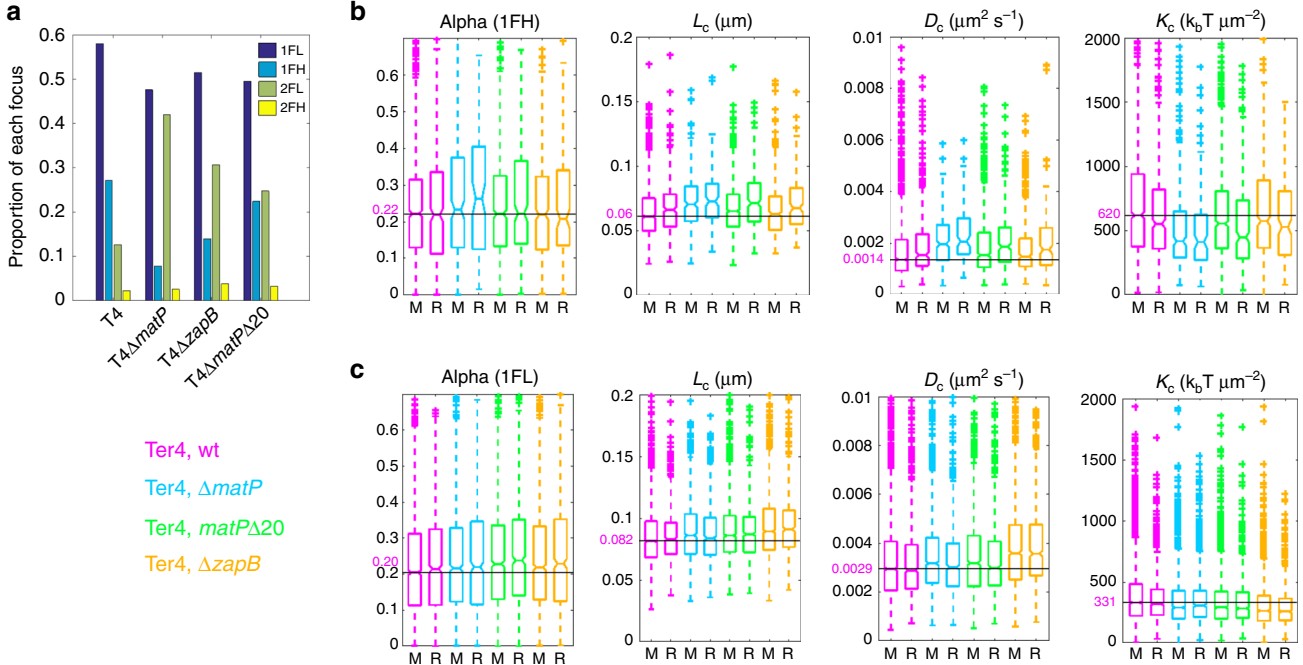

**Fig. 3 Ter4 foci are more mobile in *matP* mutants. a** Proportion of each type of focus in Ter4 mutant strains. The four parameters $\alpha$, $L_c$, $D_c$ and $K_c$ were calculated for 1FH Ter4 foci (**b**), and 1FL Ter4 foci (**c**). Wild-type (pink) and mutant backgrounds ($\Delta matP$, blue; $matP\Delta20$, green; $\Delta zapB$, orange) are presented as in Fig. 2d. To help comparing the values, a line has been drawn at the value for the foci at mid-cell in the wt background, the value indicated in pink, and the outliners at high values have been cut off. Box plots show the median of the distribution, the 25th and 75th percentiles of the samples (respectively the bottom and top of the box), the lowest and top values within a range of 1.5 times the interquartile range (dotted lines), and outliers of these as crosses. The notches display the variability of the median; non-overlapping notches between samples indicate different value of the medians at a 5% significance level. **b** Ter4 wt, $n = 1374$ for M, $n = 412$ for R; Ter4$\Delta matP$, $n = 258$ for M, $n = 115$ for R; Ter4$matP\Delta20$, $n = 968$ for M, $n = 267$ for R; Ter4$\Delta zapB$, $n = 475$ for M, $n = 174$ for R. **c** Ter4 wt, $n = 2744$ for M, $n = 1064$ for R; Ter4$\Delta matP$, $n = 1057$ for M, $n = 754$ for R; Ter4$matP\Delta20$, $n = 1805$ for M, $n = 918$ for R; Ter4$\Delta zapB$, $n = 1598$ for M, $n = 786$ for R. Source data are provided as a Source Data file.

Ter4 foci (Fig. 3a). The number of cells with two FL increased, whereas those with a single FH decreased. This effect was slightly more pronounced in the $\Delta(zapB)$ than in the $matP\Delta20$ strain. This shows that two activities of MatP are mainly required for FH formation: one not affected in MatP$\Delta20$ and the other one depending on an interaction with ZapB.

A detailed analysis of Ter4 FH in the two mutants was fully consistent with the above conclusion (Fig. 3b, Supplementary Data 3). FH confinement decreased and mobility increased (higher $L_c$ and $D_c$, lower $K_c$) compared to the wild-type strain; these effects were more marked in the $matP\Delta20$ strain, yet lower than in the $\Delta(matP)$ strain. In both mutant strains, FH were more confined and less mobile when located at mid-cell, as in the wild-type strain.

In the $matP\Delta20$ strain, Ter4 FL behaved as in the $\Delta(matP)$ strain (Fig. 3c), i.e., slightly increased $L_c$ and $D_c$ and decreased $K_c$ compared to the wild-type strain, indicating a moderate decrease of confinement and increase in mobility. MatP activities other than tetramerisation and/or interaction with ZapB are thus not required to constrain FL. Surprisingly, the $\Delta(zapB)$ mutation had a larger effect than either the $\Delta(matP)$ or the $matP\Delta20$ mutation on Ter4 FL (Fig. 3c). This was unexpected but may be explained by properties of ZapB that are independent of its interaction with MatP (see "Discussion").

**Only one MatP activity is mediated by TopoIV control.** Since MatP was proposed to control the removal of catenanes by TopoIV[15], we assayed the effect of a moderate overproduction of both TopoIV subunits. This was previously reported to shorten the post-replicative co-localisation period of non-*ter* loci by

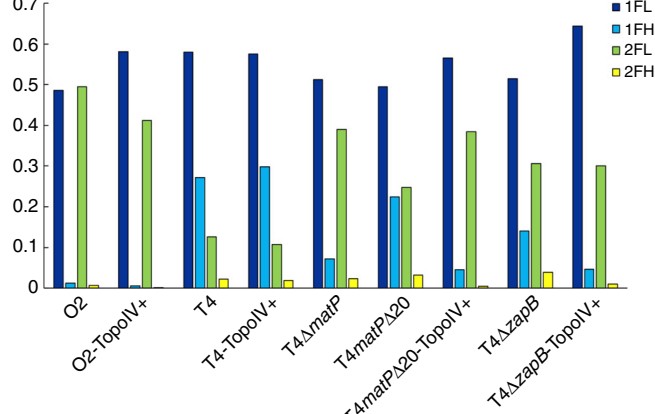

**Fig. 4 An increased amount of TopoIV decreases the number of FH in mutant strains.** Proportion of each type of focus for Ori2 (O2) or Ter4 (T4) strains. The TopoIV was slightly overproduced in strains marked as TopoIV+. Source data are provided as a Source Data file.

premature resolution of precatenanes[37]. Overproduction of TopoIV increased the portion of cells with a single FL, suggesting it somehow modifies the cell cycle, delaying replication and/or postponing cell division (Fig. 4). As expected, the number of Ori2 FH decreased (from 1.3 to 0.5%), consistent again with most FH being paired loci. Strikingly, this effect was not observed at Ter4 in the wild-type strain but was clearly observed in both the

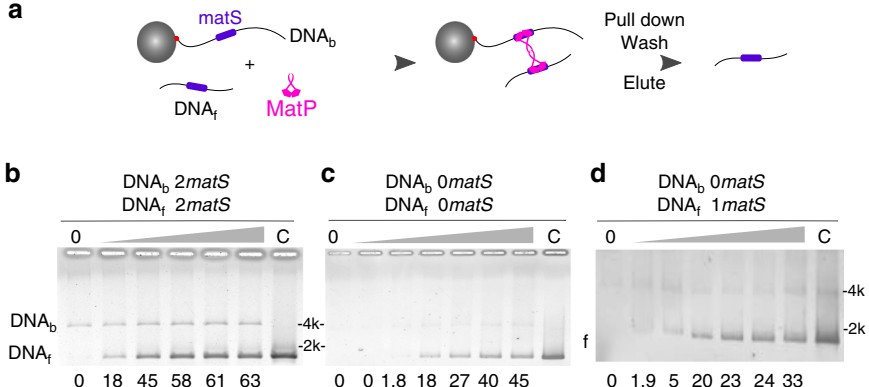

**Fig. 5 MatP can bridge DNA. a** Scheme of the pull-down experiment. A biotinylated DNA (DNA$_b$), containing 0, 1 or 2 *matS*, is attached to a streptavidin-covered, magnetic bead. Free DNA (DNA$_f$), containing 0, 1 or 2 *matS*, is added to the reaction, alongside a competitor (40 bp, non-specific, double-stranded DNA). Different concentrations of MatP are added to the mix. The reactions are pulled-down a first time, rinsed with buffer, eluted with 0.1% SDS, which denatures MatP but not the streptavidin, and pulled-down again. DNA$_f$ is recovered if MatP has induced bridging. Those are then loaded on an agarose gel. **b–d** Examples of gels obtained with both DNA$_b$ and DNA$_f$ containing 2 *matS* (**b**), no *matS* (**c**) or 1 *matS* on DNA$_f$ (**d**). Concentration of MatP is from left to right: 0, 0.1, 0.25, 0.5, 1 and 2 μM, with an additional reaction with 4 μM for (**c**). The percentage of recovered DNA$_f$ is shown below each lane; lane C shows the initial amount of DNA$_f$ (20 ng). Quantitation can vary from gel to gel but the trend is the same within three independent experiments. The type of DNA observed on the gel is indicated on the left of the gel (f for DNA$_f$). Molecular weight markers are indicated on the other side, in kbp.

*matPΔ20* and Δ(*zapB*) strains. There, the rate of Ter4 FH dropped to the one measured in the Δ(*matP*) strain (about 4%). We conclude that post-replicative co-localisation between sister *ter* loci results from at least two different mechanisms involving MatP: one depends on the last 20 C-terminal residues of MatP and on ZapB and is not affected by increased levels of TopoIV, the second one depends on an increased level of TopoIV, suggesting a MatP-dependent catenane persistence.

**MatP bridges DNA molecules in vitro.** As we assume FH contain paired loci dependent on MatP, we attempted to characterise DNA bridging by MatP. We designed an assay based on DNA pull-down (Fig. 5a). A biotinylated DNA molecule containing 0–2 *matS* was bound to a streptavidin-covered magnetic bead (DNA$_b$). This complex was mixed with a free DNA (DNA$_f$) containing 0 or 2 *matS*. The mix was then incubated with MatP, washed, pulled-down with a magnet and eluted with a low concentration of SDS. The amount of DNA$_f$ recovered represents the capacity of MatP to bridge the two DNA molecules. When they both contained two *matS* (Fig. 5b), 18% of DNA$_f$ were recovered with the lowest concentration of MatP used (0.1 μM), increasing to 63% at the highest concentration (2 μM). When repeating the same experiment with DNA that did not contain *matS*, DNA$_f$ was recovered with lower efficiency (Fig. 5c). When only one DNA contained a *matS*, the amount of DNA$_f$ recovered was intermediate (Fig. 5d and Supplementary Fig. 6). We conclude that MatP is able to bridge independent DNA molecules containing or devoid of *matS*. The presence of *matS* stimulates this activity and/or stabilises the complexes formed.

**MatP bridging activity involves non-specific DNA binding.** To better describe MatP bridging activity, we used a high-throughput tethered particle motion (TPM) set-up[38]. This tracks beads attached at one end of a DNA molecule while the other extremity is tethered to a coverslip (Fig. 6a). The amplitude of motion at equilibrium of the bead ($A_{eq}$) directly depends on the apparent length of the DNA[39]. We used DNA containing 0, 1 or 2 *matS* separated by 1207 bp. The traces (example Fig. 6b) are plotted as densities of probability of their $A_{eq}$ and fitted to Gaussian distributions (Fig. 6c). Without protein we observed a single

population centred on 250 nm (Fig. 6c, *). Adding MatP resulted in the displacement of the whole population toward shorter $A_{eq}$, independently of the DNA used. This corresponds to an apparent shortening of around 30 nm (Fig. 6c, **). This moderate decrease in $A_{eq}$ cannot correspond to MatP-induced DNA looping between the *matS* sites, because it is independent on *matS*. Moreover, the shortening predicted from bridging two *matS* would be around 100 nm. An equivalent moderate decrease in $A_{eq}$ was previously observed using another site-specific, DNA-binding protein in the same set-up and was attributed to protein binding to a single site[40,41]. Surprisingly, this moderate decrease in $A_{eq}$ was not observed with MatPΔ20 (Fig. 6c). We verified that MatPΔ20 and MatP bound *matS*-containing DNA (Supplementary Fig. 7a). Tagged or untagged versions of MatPΔ20 bound a matS-containing DNA with equivalent efficiency as MatP. However, the mobility of the complexes was unexpected. Indeed, MatPΔ20-tag formed complexes migrating faster than MatP, despite being twice bigger (Fig. 7). Assuming MatPΔ20 binds DNA as a dimer[12], these observations suggest that MatP binds DNA as a tetramer. This explains the difference observed between MatP and MatPΔ20 in TPM experiments, suggesting the moderate decrease in $A_{eq}$ (Fig. 6c, **) is due to tetrameric MatP binding to a single *matS*, or to non-specific DNA, and changing DNA conformation.

A second population with shorter $A_{eq}$ was obtained upon incubation of the 2-*matS* DNA with MatP (Fig. 6c, ***). Gaussian fitting of this population indicates that it was centred at 147 nm (±22), corresponding to the 100 nm shortening predicted for looping between two *matS*. Accordingly, this peak was neither observed when using DNA without *matS* nor when using MatPΔ20 (Fig. 6c). However, it was also detected upon incubation of the 1-*matS* DNA with MatP. This peak was centred on 153 nm (±23) and contained three times fewer events (13%) than with the 2-*matS* DNA (36%). The second peak formed with DNA containing one or two *matS* exhibited very similar centres, as if 1200 bp was the most favourable inter-*matS* distance for looping by MatP. We next repeated the experiment using a 2-*matS* DNA with non-specific DNA as a competitor. The moderate decrease in $A_{eq}$ was readily observed (Supplementary Fig. 8a; **), but the peak corresponding to a greater shortening was not, showing that pairing of distant DNA loci by MatP is

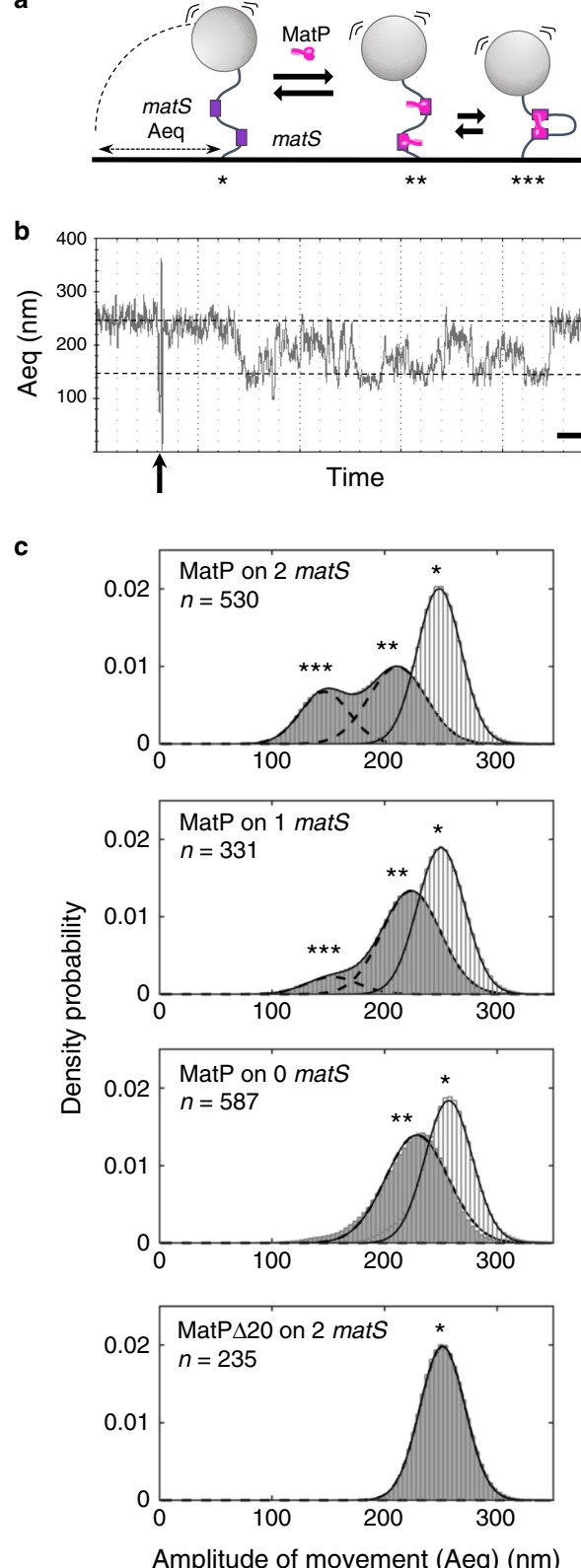

**Fig. 6 MatP can loop DNA. a** Scheme of the TPM set-up. A glass coverslip is coated with PEG and neutravidin. A DNA molecule is attached to that surface by a biotin bound to one of its 5′ end. A latex bead coated with anti-digoxigenin is bound to the other extremity of the DNA molecule by the presence of digoxigenin on this 5′ end. The amplitude of the Brownian motion of the bead ($A_{eq}$) depends on the size of the DNA molecule that tethers the bead to the glass surface (for details see text and "Methods"). Right: if MatP joins the 2 *matS* sites together, the $A_{eq}$ decreases. **b** Example of a track following a bead tethered to a 2-*matS* DNA as a function of time. The dotted lines indicate the expected $A_{eq}$ for a naked DNA (top) or looped DNA (bottom). MatP is added after 2 min (arrow) and the bead tracked for another 13 min. Time scale is 1 min (bar). **c** Probability distributions of $A_{eq}$, before protein injection (light grey histogram), or during the 5 min following the injection (dark grey histograms). The type of DNA (0–2 *matS*), the type of MatP (wt or Δ20) and the number of tracks obtained for each condition are indicated on the graphs. MatP: *population centred on (mean ± SD, in nm): 248 ± 19; 250 ± 21 and 257 ± 21 for the DNA containing 2, 1 and no *matS* site, respectively. **population centred on (mean ± SD, in nm): 211 ± 25, 224 ± 26 and 229 ± 28 for the DNA containing 2, 1 and no *matS* site, respectively. ***population centred on (mean ± SD, in nm): 147 ± 22 and 153 ± 23 for the DNA containing 2 and 1 *matS* site, respectively. MatPΔ20: *population centred on (mean ± SD): 252 ± 20 nm. Source data are provided as a Source Data file.

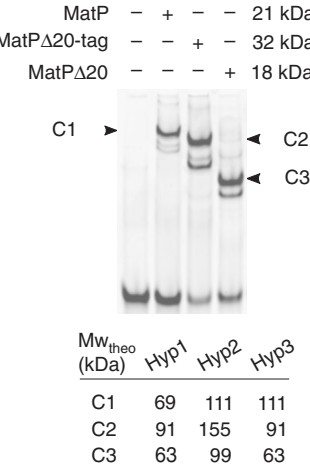

| Mw$_{theo}$ (kDa) | Hyp1 | Hyp2 | Hyp3 |
|---|---|---|---|
| C1 | 69 | 111 | 111 |
| C2 | 91 | 155 | 91 |
| C3 | 63 | 99 | 63 |

**Fig. 7 MatP binds *matS* as a tetramer.** EMSA experiment showing the interaction between a 41 bp DNA fragment containing *matS* (MW$_{theo}$ = 27 kDa) and 3 μM of MatP (MW$_{theo}$ = 21 kDa), his-tagged MatPΔ20 (MW$_{theo}$ = 32 kDa), or untagged MatPΔ20 (MW$_{theo}$ = 18 kDa). Based on the theoretical molecular weight (MW$_{theo}$) of the DNA molecule and of the different proteins used, we have estimated the theoretical molecular weight of the different protein-DNA complexes following three different hypotheses. Hypothesis 1 (Hyp1) proposes that MatP and MatPΔ20 bind DNA as a dimer; hypothesis 2 (Hyp2) proposes that MatP and MatPΔ20 bind DNA as a tetramer; hypothesis 3 (Hyp3) proposes that MatP binds DNA as a tetramer, whereas MatPΔ20 binds DNA as a dimer. Note that only hypothesis 3 proposes theoretical molecular weights that are in accordance with EMSA results.

sensitive to non-specific DNA competition. The analysis of the kinetics of loop formation (Supplementary Fig. 8b, c) showed that the presence of *matS* enhances the formation of loops by increasing the duration of long-lived looped events.

Taken together, these results show that MatP can pair distant DNA loci in TPM. This activity depends on the 20 C-terminal

residues of MatP, is stimulated by the presence of *matS* but is highly dynamic, and strongly sensitive to the presence of non-specific DNA.

This prompted us to analyse in more details the MatP/*matS* complexes. Using two DNA probes carrying a single *matS* and labelled with different fluorophores, we performed an EMSA (Supplementary Fig. 7c). We failed to detect complexes containing both fluorophores, showing that the MatP/DNA complexes contain a single DNA molecule. This suggests that when MatP

tetramers are bound to *matS*, one DNA-binding domain is involved in the specific interaction with *matS*, while the second is free. This *matS*-free dimer could then interact with non-specific DNA and/or with other proteins, including ZapB. We directly assayed this last hypothesis by adding purified ZapB in our in vitro assays, but failed to detect significant effects. ZapB did not change the pattern obtained in EMSA (Supplementary Fig. 9), nor the quantity of DNA$_f$ retrieved in pull-down (Supplementary Fig. 10). Thus, if the *matS*-bound MatP tetramer interacts with ZapB, this interaction is too weak, transient or unstable to be detected in our experiments.

## Discussion

We combined different approaches to investigate the organisation of the *ter* region and the role of MatP. These results yield several new observations and unexpected conclusions: (1) *ter* loci are not intrinsically less mobile than *ori* loci; (2) a significant proportion of *ter* foci show a high fluorescence, preferentially when they are located at mid-cell, and are much less mobile; (3) the proportion of these foci and their low mobility depend on all described activities of MatP, specifying two separable processes, of which one depends on the persistence of catenanes, while the other one depends on interaction with ZapB; (4) MatP binds *matS* as a tetramer but this complex contains only one DNA fragment; (5) the bridging of *matS* sites can be observed in vitro but is efficiently competed by non-specific DNA. From these observations, we conclude that MatP constrains *ter* mobility only at a specific stage of the cell cycle, when replication has terminated and sister chromosomes are paired by their *ter* regions (D period of the cell cycle). MatP does so via at least two activities, only one depending on its C-terminal domain. This last activity certainly depends on tetramerisation, but not on bridging remote *matS*. Indeed, the pairing of *matS* can be observed in vitro but is readily challenged by non-specific DNA and is thus likely irrelevant in vivo.

Loci of the *ter* region have been reported to be less mobile than other regions[9,12,14,21]. However, these comparisons did not consider the context of the foci, particularly their intensity. Tracking foci over short-time intervals, focusing on local DNA constraints, has revealed that mobility depends on foci intensity and on their sub-cellular positioning[27]. Our data extend this observation and further show the importance of the labelling system. Indeed, using the less invasive pMT1-derived system, significant differences in mobility between an *ori*-proximal and a *ter*-proximal locus are observed only for the most intense foci (FH). Indeed, less intense foci (FL) of the *ter* and *ori* loci show the same decrease in mobility when intensity increases (Fig. 1). The trajectories of FL revealed slight but significant differences between *ori* and *ter*, including a decreased $\alpha$ anomalous coefficient, suggesting *ori* is more condensed than *ter* (Fig. 2). This may be consistent with recent images of the *E. coli* nucleoid showing a donut shape in round cells, in which the *ter* region appears less condensed than the rest of the chromosome in a MatP-dependent way[42]. Higher $\alpha$ coefficient may be linked to the MatP-mediated exclusion of MukBEF from *ter*[13,15,16]. However, the difference in $\alpha$ we measured is not MatP-dependent (compare Figs. 2 and 3), arguing against this hypothesis. Despite its higher condensation, the *ori* locus appears freer to diffuse than the *ter* locus, as shown by its higher diffusion coefficient (Fig. 2). This difference is only partly suppressed by a mutation of *matP* and better suppressed by a mutation of *zapB* (compare FL foci in Figs. 2 and 3). Since ZapB self-assembles into large structures and clusters around *ter* (in a MatP-dependent manner) and the divisome (in a MatP-independent manner)[17,36,43,44], we suspect these clusters limit the diffusion of *ter* loci via an interaction with *matS*-bound MatP but also by a MatP-independent mechanism, yet to be described.

Highly fluorescent foci of the *ter* locus (Ter4 FH) show very distinctive properties. They are less mobile than expected from their intensity compared to the *ori* locus (Fig.1). They are preferentially localised at mid-cell and depend on MatP, TopoIV and ZapB (Figs. 2–4). They are in majority single foci and, in mutant strains, their decrease correlates with an increase in two-foci cells (Fig. 3). From these observations, we conclude that most FH contain pairs of sister loci. An estimation of cell cycle periods duration agrees with this view. In our strain and growth conditions, about two-third of the cells have completed replication[45]. Assuming *ter* segregates at the onset of cell constriction[10] and that about 25% of the cells are constricting in a growing population, the 27% FH we observed can be restricted to cells having completed replication and not initiated constriction. Detailed analysis shows that the $\alpha$ coefficients of FH and FL are close for Ter4, suggesting paired sister *ter* regions are not more condensed than single ones. Ter4 FH also show lower $L_c$ and $D_c$ and higher $K_c$ than FL, consistent with their very low mobility and high constraint. Surprisingly, this is only partly suppressed by a mutation of *matP* (compare Figs. 2 and 3) and even less by *matP*$\Delta$*20* or a mutation of *zapB* (Fig. 3). These results agree with the low mobility of Ter4 FH being primarily due to the post-replicative pairing of sister *ter* regions. The role of MatP would therefore be to delay the separation of sister *ter* regions, hence its drastic effect on the frequency of FH, but only marginally to constrain their mobility *per se*.

At least two activities of MatP are involved in post-replicative pairing; one, requiring the 20 C-terminal residues of MatP, depends on ZapB but not on catenanes and the other one, independent on the last 20 residues, depends on catenanes but not on ZapB—assuming that over-expressing TopoIV leads to catenanes removal. The pairing activity unaffected in *matP*$\Delta$*20* involves the control of sister chromosome decatenation since it is suppressed by overproduction of TopoIV (Fig. 4). Consistently, inhibition of decatenation by depleting active TopoIV hinders sister chromosome separation[37,46–49]. MatP may control the activity of TopoIV by several mechanisms. First, MatP could exclude TopoIV from *ter* by excluding MukB[13,15,16] through direct interactions[15,50,51]. Second, catalysis by TopoIV at the *dif* site decreases in a $\Delta$(*matP*) strain[11]. Third, FtsK, which also activates TopoIV[52,53], segregates sister *ter* regions in a MatP-dependent manner[10]. Besides, the functions of MatP depending on its last 20 residues may involve the formation of tetramers and depends on the interaction MatP-ZapB. Indeed, most of the effect of the *matP*$\Delta$*20* mutation on pairing depends on *zapB* (Fig. 3). The *ter*-linkage, via ZapB and ZapA, thus appears required for the maintenance of a normal pairing of the *ter* regions, which suggests that it is partly independent of their catenation. Lastly, a small but significant part of MatP activity dependent on its C-terminus is independent on ZapB (Fig. 3). Thus, either the last 20 residues participate in an undescribed activity of MatP or MatP tetramerisation plays some role in pairing *ter* independently of either MukB or ZapB.

We further characterised the interaction of MatP with DNA and its tetramerisation. Our data are fully consistent with MatP forming tetramers when it interacts with *matS* (Fig. 7), as previously reported[12]. However, despite dedicated experiments, we detected only one DNA fragment per tetramer (Supplementary Fig. 7), but could see pairing of DNA molecules in vitro in two assays. This pairing activity only poorly depended on the presence of *matS* and was readily challenged by non-specific DNA (Figs. 5 and 6). We conclude that *matS*-*matS* looping or bridging by MatP is unlikely to occur frequently or stably in vivo, consistent with their absence in contact maps[13]. MatP tetramers may instead pair *matS* with non-specific DNA, ensuring a part of *ter* pairing this way. This

should lead to compaction of the *ter* region, which is argued against by the absence of anomalous component modification in Δ(*matP*) (Fig. 3). In addition, chromatin immunoprecipitation of MatP did not reveal non-specific binding of MatP around *matS* sites[13]. An attractive model would thus be that MatP tetramers specifically serve to bridge *matS*-containing DNA with ZapB, excluding other binding activity, so that the DNA-bridging activity of MatP would be observed only when ZapB is absent. While multiple findings point to the existence of a Ter/MatP/ZapB/ZapA/FtsZ complex in vivo[14,17,43,44,54], our attempts to detect such an interaction in vitro (Supplementary Figs. 9 and 10) were unsuccessful. These negative results do not rule out the model and could be explained by the complexes being only transients or requiring other actors like ZapA and/or FtsZ.

Taken together, our data support the view that MatP mainly acts to pair sister *ter* regions until the onset of cell division but has little effect on their dynamics when unpaired. To do so, tetramers of MatP bind *matS* sites and act in at least two ways, which can be genetically separated. This globally results in delaying decatenation by TopoIV until FtsK gets activated[55–58] and segregate the *ter* regions by promoting both decatenation and dimer resolution at the *dif* site[10,11].

## Methods

**Strains, media, plasmids.** *E. coli* strains were all derived from MG1655 and provided by Espeli[21]. Briefly, *parS* sequences were inserted at positions 4197685 bp for Ori2 locus (*parS*-P1 and *parS*-pMT1), 1395706 bp for Ter3 (*parS*-P1), and 1444252 bp for Ter4 (*parS*-pMT1) loci. Strains carrying a parS-P1 sequence were transformed with pALA2705, and strains carrying *parS*-pMT1 were transformed with pFHC2973[28,29]. The Δ(*matP*) and Δ(*zapB*) deletions were transferred by P1 transduction from strains JW939 and JW3899 of the Keio collection[59]. The *matPΔ20* strain was obtained from Boccard[12]; the mutant gene was transduced into the *parS*-pMT1 labelled strains. Ampicillin (50 μg/mL), kanamycin (25 μg/mL), chloramphenicol (10 μg/mL) or spectinomycin (20 μg/mL) were added when needed. To overexpress TopoIV, strains were transformed with a pWX35[37] containing the spectinomycin resistance. The leakage from the arabinose promoter was sufficient to observe increased decatenation.

**Microscopy measurements.** Strains were grown at 30 °C in M9 broth (Difco) supplemented with complementary salts ($Mg_2SO_4$ 2 mM, $CaCl_2$ 100 μM, tryptophan 4 μg mL$^{-1}$ and thymidine 5 μg mL$^{-1}$), glucose (0.4%) and CAA (0.1%) for 12 h, then diluted 2000× in fresh M9-glucose (0.4%). At an $OD_{600nm} \approx 0.1$, ParB-pMT1 fusion proteins production was induced for 30 min with 30 μM IPTG. Cells were then deposited on a 1.5% agar pad in M9-glucose, incubated for 2h30 (two cell cycles) at 30 °C, and imaged. A control experiment was done with fixed cells, which were grown as above, centrifuged, resuspended in a solution of 2% paraformaldehyde in PBS (Bioclear), incubated at 4 °C for 30 min and imaged as follows (Supplementary Fig. 11).

Imaging was done as previously described[27]. Briefly, movies were taken on a Nikon Eclipse TiE with a 60× oil-immersion objective; the images were further magnified with a 2.5× TV adaptor before detection on an Andor iXon EM-CCD camera. Imaging was done at 2 fps with a 0.1 s exposure, for 20 s to avoid photobleaching.

**Image analysis and loci tracking.** We achieved high precision localisation of foci on each frame by two-dimensional fitting of a Gaussian function to the diffraction limited intensity distributions of individual loci[27]. This has a higher precision than typical displacements between successive frames. Particle tracks can then be obtained by matching the nearest objects in successive frames. The centre of mass motion of all the common loci in the image pair is subtracted to remove collective motion related to microscope vibration. Loci have a distribution of initial intensity, and undergo photobleaching. Tracks were analysed using a custom written Matlab R2018a script, as previously described[27]. This was further adapted for the shorter trajectories of 40 images in our experiments. Briefly, the tracking consists of three main steps: (1) localisation of candidate particles. The aim of this step is to obtain an estimate of the particle localisation; (2) subpixel resolution detection of the position. Using the original unprocessed images, the regions around the candidate particles are fitted to a 2D Gaussian; (3) linking of the trajectories. In this step, the positions detected along the different time frames are assembled to reconstruct the particles trajectories.

**Analysis of trajectories to extract the mobility parameters.** The parameters have been described by Amitai et al.[32], and they were extracted using the same

algorithm. These parameters provide independent, complementary information on first and second moment statistics:

(1) The length of constraint $L_c$ is defined as the SD of the locus position with respect to its mean averaged over time. This parameter provides estimation for the apparent radius of the volume explored by a finite trajectory. For a trajectory containing $N_p$ points, where

$$R_c = (k\Delta t) \tag{1}$$

is the position of a locus at a time $t$, $L_c$ is obtained from the empirical estimation:

$$L_c = \sqrt{\text{Var}(R_c)} = \sqrt{\frac{1}{N_p}\sum_{k=1}^{N_p}(R_c(k\Delta t) - R_c)^2}. \tag{2}$$

It characterises the confinement of a locus, which in other studies has been reported as the radius of confinement ($R_{conf}$—not to be confused with $R_c$). The $R_{conf}$ is computed from the asymptotic plateau of a mean square displacement (MSD) curve, and is therefore limited to trajectories that plateau. This is strongly influenced by the length of image acquisition. The advantage of computing $L_C$ is that it gives a robust estimate of the volume:

$$V = \frac{4}{3}\pi L_c^3, \tag{3}$$

occupied by the trajectory and can be used on any kind of trajectory, as it does not require a plateau.

(2) The anomalous exponent $\alpha$ is computed from the MSD behaviour for small increments:

$$C(t) = (R_c(\tau + t) - R_c(\tau))^2 \approx t^\alpha \tag{4}$$

$\alpha$ was estimated by fitting the first six points of the MSD of an SPT by a power law $t^\alpha$.

(3) The effective spring coefficient $K_c$. The spring force acting at position $x_a$ and measured at position $x_m$ is represented by:

$$\mathbf{F} = -K_c(x_m - x_a). \tag{5}$$

The spring constant $K_c$ allows us to estimate the effect of local tethering interactions around the locus of interest[60]. This tethering can arise from interactions of the locus with other chromosomes or cellular substructures, such as the membrane. These interactions cannot be measured directly but they can be inferred from SPTs.

(4) The effective diffusion coefficient $D_c$ reflects the second-order statistical properties of a trajectory. This diffusion coefficient accounts for local crowding that may vary along the trajectory.

**Calibration of the foci.** In order to estimate the copy number of fluorophores, we implemented a custom intensity calibration method in MATLAB (R2018a), based on the principle described in ref. [61]. The calibration method aims at estimating the ratio between the intensity of foci and the number of GFP molecules by exploiting the intrinsic fluctuations of intensity generated by the random photobleaching process. The key idea is that the variance of the intensity drop depends on the number of emitting GFP molecules contained in the focus at the beginning of acquisition, i.e., a higher number of emitting GFP molecules corresponds to a smaller variance in the relative intensity loss (see below). We estimate the intensity/copy number ratio (calibration ratio) by binning the foci by initial intensity, and evaluating the dependence of the variance of the intensity drop on the bin.

**Detailed calculation.** Given a focus with n emitting molecules, the observed intensity will be:

$$I = \nu n. \tag{6}$$

Here we aim at estimating the calibration ratio $\nu$ from the fluctuations in the intensity drop due to photobleaching.

At any given time $t$, the number of emitting GFP molecules in a focus with $n_0$ initial emitting molecules, is described by a binomial distribution with $n = n_0$ and $p = e^{\frac{-t}{\tau}}$ where $\tau$ is the bleaching time constant. The variance of the binomial distribution is given by:

$$\text{Var}(\Delta n) = n_0 p(1 - p), \tag{7}$$

where $\Delta n$ represents the number of bleached molecules, and $p$ the probability of bleaching until time $t$. Then the contribution of bleaching to the variance of the intensity drop is:

$$\text{Var}(\Delta I)_b = \nu^2 n_0 p(1 - p) = \nu I_0 p(1 - p), \tag{8}$$

where $\Delta I$ represents the total intensity loss for a given focus and the subscript b indicates that the term is relative to the component of variance coming from fluctuations in the bleaching process.

This establishes a linear relationship between the variance of the intensity drop and the calibration ratio that can be estimated as the slope of $\text{Var}(\Delta I)_b$ as a function

of $I_0 p(1 - p)$ on a linear fit (with intercept constrained at 0) on several initial intensity bins (Supplementary Fig. 3c). $P$ is estimated from the data through a fit on $\Delta I$ as a function of $I_0$ (Supplementary Fig. 3a). This also allows to estimate the bleaching time constant $\tau$ that will be used in the following steps.

In order to estimate the contribution of bleaching to the total intensity variance, we model it as follows:

$$\text{Var}(\Delta I)_{\text{total}} = \text{Var}(\Delta I)_b + \text{Var}(\Delta I)_{\text{other}} = \nu I_0 p(1 - p) + \gamma I, \quad (9)$$

where $\gamma I$ groups non-bleaching variance contributions such as shot noise, and is expected to be proportional to intensity.

Since we expect $p(t) = e^{\frac{-t}{\tau}}$ and $I(t) = I_0\, e^{\frac{-t}{\tau}}$ we can substitute it in Eq. (9) and fit the following expression to the observed variance as a function of time (Supplementary Fig. 3b):

$$\text{Var}(\Delta I)_{\text{total}} = \gamma e^{\frac{-t}{\tau}} + \nu I_0 \left( e^{\frac{-t}{\tau}} - e^{-2\frac{t}{\tau}} \right). \quad (10)$$

The bleaching dependent part of variance (the second term of the expression) has a different functional dependence on time, and therefore can be disentangled from other sources of variance and used to estimate the calibration parameter through the above described linear fitting. This process also allowed to verify that the bleaching contribution is indeed the dominant contribution in the empirical variance. As an additional pre-processing step, we estimate and subtract background intensity by extrapolating the intensity at which no bleaching is observed (Supplementary Fig. 3a).

**Statistics and reproducibility**. Each strain was tested at least three times independently, and the data were pooled in the same data file after checking that the results obtained with each replicate were comparable. The final numbers of foci used for data analysis are indicated in Supplementary Data 1 and in the legends of the figures. Supplementary Data 2 shows the median for each dataset.

To compare distributions, a two-sample Kolmogorov–Smirnov test was used and $p$ values are indicated in the text and Supplementary Data 3. Unless stated, the distributions of foci at mid-cell (M) were used to compare two strains or type of foci (FH or FL).

**MatP purification**. A pET15b containing *matP* or *matP∆20* was transformed into BL21DE3 cells. After 2 h induction with 1 mM IPTG, cells were centrifuged and pellets were resuspended in RB1 (20 mM Tris pH 7.5, 300 mM NaCl, 1 mM DTT, 5% glycerol and protease inhibitor (complete EDTA-Free, Roche)). Cells were lysed by sonication, centrifuged and resuspended in RB1 for a step of ultra-centrifugation (50,000 rpm for 90 min at 4 °C). The supernatant was loaded onto a heparin column (Hi-trap Heparin HP, GE Healthcare Life Sciences) and MatP was eluted with an NaCl gradient (0.3–1 M) in RB1 with 5 mM MgCl2. MatP fractions were pooled, dialysed (20 mM Tris pH 7.5, 250 mM NaCl, 1 mM DTT, 5 mM MgCl2, 20% glycerol) and frozen.

His-tagged MatP∆20 was cloned and expressed as above; after the first centrifugation, it was loaded onto a HisTrap column (HisTrap, GE Healthcare Life Sciences) and eluted with an imidazole gradient (0–0.5 M; 10 column volumes). MatP∆20 fractions were pooled, dialysed (20 mM Tris pH 7.5, 250 mM NaCl, 1 mM DTT, 2 mM EDTA, 10% glycerol) and frozen.

**Pull-down experiments**. Magnetic beads (1 µL/reaction, Streptavidine Magne-Sphere® Paramagnetic Particles, Promega) were washed twice in PBS and once in RB2 (20 mM Tris pH 7.5, 1 mM MgCl2, 150 mM NaCl, 1 mg mL⁻¹ BSA), then resuspended in RB2. Biotinylated DNA$_b$ (10 ng) (DNA$_{b(2matS)}$: 3746 pb (from PCR run with OEB3-4), DNA$_{b(1matS)}$: 3800 pb (OEB5-6), DNA$_{b(0matS)}$: 3700 pb (OEB7-8)) was added and incubated at RT for 30 min. The DNA-bead complexes were then washed and resuspended in RB2, and the non-biotinylated, DNA$_f$ (20 ng) was added (DNA$_{f(2matS)}$: 1701 pb (OEB9-10), DNA$_{f(1matS)}$: 1717 pb (OEB5-11), DNA$_{f(0matS)}$: 1688 pb (OEB7-12)), along with 100 nM competitor DNA (25 bp, double-stranded, non-specific oligo). Finally, MatP (respectively 0, 0.1, 0.2, 0.5, 1 and 2 µM) was added and reactions were incubated for 15 min at RT. After this time, the supernatant was removed and reactions were quickly washed with 15 µL RB2, then resuspended in 15 µL RB2 + 0.1% SDS. After 10 min, the supernatant was deposited on a 0.8% agarose gel. DNA was visualised with Sybr-Green (Life technologies).

**Multiplexed tethered particle motion (TPM)**. The overall TPM procedure, including data analysis, has been described previously[38,62]. DNAs were obtained as follows: PCR with F2060 and R2016 on pBR328 for 0*matS*-DNA (2588 bp); PCR with F2060 and R2016 on pTOC7 for 1*matS*-DNA (2506 bp); PCR with F1201 and R1201 on pTOC6 for 2*matS*-DNA (2443 bp). Fifty picomolar DNA was incubated with 50 pM anti-digoxigenin-covered beads (antibody: Roche, #11093274910 and carboxylated-modified beads F1-XC030, Merck-Estapor) in Buffer A (PBS 1X, 1 mg mL⁻¹ Pluronic F-127, 0.1 mg mL⁻¹ BSA, 0.1% Tween 20 and 0.05% Triton 100X) for 20 min at RT. The complexed beads-DNA were incubated O/N in the chambers at 4 °C in RB3 (PBS 1X, 5 mg mL⁻¹ Pluronic F-127, 0.1 mg mL⁻¹ BSA, 0.1% SDS and 0.05% Triton 100X) and the free beads were washed with RB3. Chambers were then washed with RB4 (20 mM Tris pH 7.5, 1 mM MgCl2, 150 mM NaCl,

0.1 mg ml⁻¹ BSA, 5 mg mL⁻¹ Pluronic F-127), and imaged for 2 min before injection of 600 nM MatP or MatP∆20. Traces were examined one by one as described in ref. [40]. Only those with an appropriate amplitude of motion measured in the absence of proteins, regarding the calibration curve, were conserved for further analysis. We analysed the kinetics of the amplitude of motion after 5 min after injection of proteins and used for that detection methods based on thresholds defined as midways between the peak positions found in the histograms of amplitude of motion. We thus defined two states: state 1, intact DNA, and state 2, apparently looped DNA and detected them on each trace. The histograms of the state duration were fitted with two exponential decays leading to $\tau_{\text{fast}}$ and $\tau_{\text{slow}}$.

**Electromobility shift assays (EMSA)**. The *matS41* DNA (41 bp) was obtained by hybridising oligonucleotides matS41F and matS41R. The *matS237* DNA (237 bp) was obtained by PCR amplification using pEL3, a pLN135 derivative containing the *matS* sequence (our lab collection *matS* site: GTGACAGTGTCAC), as a matrix and matSF and matSR as oligonucleotides. Binding reactions were done in buffer containing 10 mM Tris (pH 7.5), 125 mM NaCl, 2.5 mM MgCl2, 0.5 mM DTT, and 5% glycerol, in the presence of 1 µM of each DNA probe, 0.25 µg of poly(dI-dC) and different concentrations of indicated proteins (3 µM of proteins in Fig. 7 and Supplementary Fig. 9, and 3 µM and 6 µM in Supplementary Fig. 7) The reactions were incubated at 30 °C for 30 min and analysed on 5% native TGE-PAGE.

**ZapB purification**. A pET32 containing *zapB* was transformed into BL21DE3 cells. After 2 h induction with 1 mM IPTG, cells were centrifuged and pellets were resuspended in Buffer A (20 mM Tris pH 7.5, 300 mM NaCl, 5 mM MgCl2). Cells were lysed by sonication, centrifuged and resuspended in RB1 for a step of ultra-centrifugation (27,000 g for 90 min at 4 °C). The supernatant was loaded onto a 1 mL HisTrap FF column (GE Healthcare Life Sciences) and ZapB was eluted with an imidazole gradient (0–0.5 M). ZapB fractions were pooled, dialysed (20 mM Tris pH 7.5, 150 mM NaCl, 5 mM MgCl2) and loaded on a Superdex 200 (GE Healthcare Life Sciences). ZapB fractions were pooled and frozen.

**Pull-down experiments with ZapB**. This experiment is performed like other pull-down experiments but with different MatP concentrations (from 1 to 6 µM) with or without ZapB (from 2 to 20 µM). Each reaction was done in double in order to be analysed by classical agarose gel electrophoresis or SDS-PAGE. For western-blot analysis, reactions were analysed by SDS-PAGE and transferred to polyvinylidene difluoride membranes (Bio-Rad) using the Trans-Blot® Turbo™ transfer system (Bio-Rad). Membranes were blocked 1 h at RT or overnight at 4 °C in 5% non-fat powder milk in PBS containing 0.05% Tween 20. Antibodies were anti-His (1:1000) (Tanaka, #631210). Blots were developed by chemiluminescence using Clarity western ECL substrate (Bio-Rad), visualised with the Chemidoc™ Touch imaging system (Bio-Rad) and analysed with Image Lab software (Bio-Rad).

**Reporting summary**. Further information on research design is available in the Nature Research Reporting Summary linked to this article.

## Data availability
The data that support the findings of this study are available from the corresponding author upon reasonable request. Source data are provided with this paper.

## Code availability
The codes used in this study to track the foci are available at https://github.com/ver228/bacteria-loci-tracker. The codes used to extract the parameters are accessible at http://bionewmetrics.org/; "Nuclear Organisation" section.

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

## Acknowledgements

We would like to thank M. Cosentino Lagomarsino, A. Javer, J. Kotar, M. Panlilio, M. Wlodarski, for useful discussions and help with experimental setups. We are also grateful to F. Boccard and O. Espeli for the kind gifts of bacterial strains and plasmids. Research in the FC group is founded by the CNRS and the Université Paul Sabatier (UPS Toulouse). Research in PC lab was supported by HFSP (RGY0070/2014) and UKRI grant EP/T002778/1.

## Author contributions

E.C., P.R., P.C. and F.C. conceived the project. E.C. performed the microscopy experiments. E.C., D.H., B.S., P.R., F.C. and P.C. discussed the microscopy results. E.C., C.T. and M.S. performed the T.P.M. experiments. E.C., M.S., P.R. and A.L. performed the biochemical experiments. T.B. designed and performed the foci calibrations. E.C., P.R., P.C. and F.C. wrote the paper.

## Competing interests

The authors declare no competing interests.
