## [Peer Review File · Nature Communications]

Reviewers' comments:

Reviewer #1 (Remarks to the Author):

The authors combined *in vivo* and *in vitro* approaches to investigate the role of MatP in organizing the terminus region. *In vivo*, they used time-lapse microscopy to monitor the mobility of *ter* region. *In vitro*, they used DNA pull down assay and tethered particle motion assay to study how MatP bridges DNA. Their major conclusions are: the terminus region is not less mobile per se, but only less mobile when replicated *ter* foci are paired; After testing ΔmatP , $\text{matP}\Delta 20$ and ΔzapB , they found that MatP's tetramerization domain, and MatP's interaction with ZapB are important for post-replicative pairing of *ter* foci. Another (unknown) activity of MatP is also required for post-replicative pairing of *ter*, and the authors proposed that this other activity could be related to MatP excluding MukB from the terminus region, which may lead to lower TopoIV at *ter* to remove precatenanes (Nolivos et al NatComm 2016). *In vitro* assays did not find MatP specifically bridging different *matS* sites, but found that MatP bridges *MatS* and non-specific DNA.

MatP is an important protein for chromosome organization and is relatively well studied. Previous work from several labs found that 1) MatP reduces the mobility of the terminus region, and was proposed to do so through bridging different *MatS* sites; 2) MatP interacts with the ZapB protein and 3) MatP excludes MukB from the terminus region. This manuscript extends previous findings and clarifies that the reduced *ter* mobility is only seen in paired foci, and proposed that MatP does not act to bridge different *MatS* sites but bridges between *matS* site and the ZapB protein. These hypotheses provide new insights into the mechanism of MatP action, but direct experimental evidence for some of these claims are missing (see below).

Major concerns:

1) A novel hypothesis for this paper is that, a tetramer of MatP does not bridge different *matS* sites, but bridge between *matS* and ZapB. Direct experimental evidence for this hypothesis is missing. There might be some experiments the authors can perform to test this. For *in vitro* experiment, perhaps the authors can use a similar experiment as in Figure 4, to use MatP bound to *matS* to pull down ZapB protein instead DNA. For *in vivo* experiment, maybe the authors could perform a ChIP-seq experiment, using antibodies against ZapB, to see whether ZapB is specifically enriched at *matS* sites, and whether the enrichment is dependent on the tetramerization domain of MatP.

2) The authors proposed that the remaining pairing activity of *ter* in $\text{MatP}\Delta 20$ involves the control of sister chromosome decatenation, that is $\text{MatP}\Delta 20$ could exclude MukB from *ter* leading to lower TopoIV levels to decatenate the replicated *ter* (Nolivos et al NatComm 2016). This hypothesis could be tested experimentally. First, the authors could mildly express TopoIV (See Wang et al GeneDev 2008; Lesterlin et al EMBOJ 2012) in WT or in $\text{MatP}\Delta 20$ to test whether *ter* pairing is reduced. Second, the authors could inactivate TopoIV by using *parCts* or *parEts* (See Wang et al GeneDev 2008; Lesterlin et al EMBOJ 2012) to test whether other loci (like *ori*) now exhibit a pairing phenomenon similar to the *ter* region.

3) The authors used *parS*-*ParB* system to visualize the *ter* region. The cells would have 1 copy of *parS* before replication and two copies of *parS* sites after replication. We would expect to see foci with roughly 1x or 2x intensity. It is thus surprising to see that the distribution of fluorescent intensity of foci is rather smooth and not bimodal (figure 1a). Could the authors comment on this?

4) Previous studies (Bates and Kleckner, Cell 2005; Wang et al, GenesDev 2008; Joshi et al PLOSGen 2013) suggest that the origin region undergoes significant cohesion (between 10-15% of the cell cycle) after replication. We would expect to see 10-15% of the cells with 2x fluorescent intensity at origin. But the authors saw only 2% of the cells with high intensity. Could the authors comment on absence of origin cohesion in this study? In fact, the mobility of the cohesed origin

foci would have been a nice control or comparison to the mobility of the paired ter foci.

5) In Fig S1, when using the pMT1 parS and P1 parS systems to label the origin, the fluorescence intensity distribution is the same for origin. However, the distribution of intensity for terminus changed dramatically. It seems that the P1 parS system is more invasive for the ter region and not invasive for the origin region. Could the authors comment on this discrepancy?

Reviewer #2 (Remarks to the Author):

Comments on Crozat et al.,

This manuscript investigates the effects of MatP on the constraints on the chromosomal ter in *E. coli*. The conclusions are (i) ter loci are not intrinsically less mobile than ori loci, even in presence of MatP; (ii) however, a significant proportion of ter foci show a more intense fluorescence, most preferentially when they are located at mid-cell and are much less mobile; (iii) the proportion of these foci and their low mobility depend on all described activities of MatP; (iv) MatP binds matS-containing DNA as a tetramer but this complex contains only one DNA fragment; (v) bridging matS-containing DNA can be observed in vitro but is efficiently competed by non-specific DNA. (copied from the discussion).

The experiments described are very carefully executed and critically analyzed, which results in a number of interesting observations that support the conclusions summarized above.

Major comments

Differences were found between the P1 and the MT1 system. The Mt1 system was chosen for further experimenting. However, why this was chosen is not very clear. P1 has more cells with one ter focus compared to P2. What was the expected number of ter foci given the mass doubling time and the DNA replication and segregation rates? The position of the P1 ter locus in the chromosome is different from the MT1 ter focus, how are you sure that they behave similarly are these foci not interfering with the binding of other proteins and the segregation of the ters? Can you provide better argumentation why the MT1 system is better?

Next the position, intensity and dynamics of the foci are analyzed from which it is concluded that highly intense foci are sister ter loci that are cohesive and constrained at midcell. The experiments were repeated in a cell deleted for MatP that forms the link between the septum and the ter regions by binding on the one hand ZapB and on the other matS on the chromosome. MatP appeared to be responsible for the confinement of the sister ters at mid cell as the number of FH decreases and the number of cells with 2 FL increases. The FH were more mobile in the absence of MatP but not in the absence of ZapB or in a Matp20 that is not able to bind ZapB or form tetramers. Interestingly in the cells with a single and double ter FL, these foci (but not the ori FL) were more dynamic and less confined in the delta ZapB than in the delta MatP strain.

In the Discussion: Since ZapB self-assembles into large structures and clusters around ter (in a MatP-dependent manner) and the divisome (in a MatP-independent manner) 36, we suspect these clusters limit the diffusion of ter loci both in presence and absence of MatP. I agree with the observed data, but how logic is your explanation? ZapB is very negatively charged. How would ZapB constrain the ter region without interacting with MatP as DNA is also negatively charged? Are we missing a factor?

In the absence of MatP, about 5% single FH are left. If I remember it correctly about 12% of the chromosomes need to be deconcatenated by FtsK/Xer. Would the most simple conclusion not be that all single FH in delta MatP are concatenated chromosomes and that MatP is not involved in this part of the ter constrains?

Next invitro experiments using DNA with 0, 1 or 2 matS sites and isolated MatP. MatP is binding in general to DNA, but with higher affinity when matS is present. It is able to bind two strands of DNA. MatP is able to loop DNA that has two matS sites. IN a TMP set up the tetrameric MatP binding to a DNA strand causes it to shorten i.e change its conformation, whereas the dimeric MatP20 has no effect on the DNA conformation. MatS is not required for these changes. MatS sites are required for looping and not sensitive to competition by matS-less DNA. But pairing of distant DNA loci by MatP is sensitive to non-specific DNA competition. Therefore, pairing of matS sites might not be relevant in vivo.

Minor comments

Introduction :

Page 4. MatP-bond sister ter . Should it not be bound?

Results:

Page 5: The MSDs obtained with the pMT1 system are higher (larger?) than with the P1 system

Page 5: unsegregated (single) foci being closer from (meaning?)mid-cell than segregated (double) ones.

Page 5: Single foci of Ter4 are located closest (closer?)to mid-cell than double foci

Figure 2C the green line is of a different color as the legend green square.

Page 6:The effective diffusion coefficient is higher for Ori2 than Ter4 (3.5×10^3 and $2.9 \times 10^3 \mu\text{m}^2/\text{s}$, respectively), showing that de-spote a higher condensation, Ori2 is freer to diffuse than Ter4. Can you give the significance of the difference?

Page 6: Compared to the values obtained for the mat locus in budding yeast ($90\text{kBT}/\mu\text{m}^2$, 32). Can you give a bit of information on the mat locus so that the reader understand why you are comparing these two loci.

Page 7: This difference might be explained by the higher intensity of FH, which should reduce their mobility 35. Assuming that a FH is 70 molecules of GFP and a LH is 33 GFP using the information of the cited paper, you can probably calculate whether the expected differences are the same as the data you observe. Is the difference between FL and FH indeed due to the increase in the number of FPs or could the interaction with FtsK play a role in a subset of the FH foci?

Page 8: This suggests that the interaction of MatP with ZapB is the main reason for FH foci formation, although the tetramerisation of MatP and a third activity that certainly involves an interaction with MukB are probably also involved. How can you have a certain third activity that probably is involved?

Figure 4b. In this figure DNAb and DNAf is observed. Apparently, also the DNA from the magnetic bead is eluted. Why is this much less the case in Fig 4c? Should the elution of the bead not be independent of MatP?

Reviewer #3 (Remarks to the Author):

The manuscript by Crozat et al. investigates the role of MatP protein in the spatial organization and temporal dynamics of the E. coli chromosome. The work shows by live cell imaging that a selected locus in the terminus region exhibits two distinct behaviours: normal mobility (similar to a control locus) and constrained mobility. The authors suggest that the constrained mobility is caused by local sister chromosome cohesion which is in large parts due to MatP protein and matS sites. DNA-DNA bridging by MatP, MatP/matS binding to ZapB and exclusion of MukBEF activity from the terminus region contribute differently to constraining the mobility of the terminus region. Finally, the authors provide biochemical experiments with purified MatP protein and DNA molecules to characterize the DNA bridging activity of MatP.

The work is executed very carefully and presented in a precise and easily accessible manner. The

main conclusions of the manuscript are well supported by the experimental data, and potential pitfalls of the observations are openly discussed. Altogether, the manuscript provides a very solid piece of work with new technical and biological insights. Few points detailed below should be considered prior to publication.

Main comments:

The characterization of different visualization systems provides valuable information for the design of future imaging experiments in different model organisms. While ParB from pMT1 seems to report the behaviour of chromosomal loci more faithfully than the widely used FROS or P1 ParB systems, it may nevertheless suffer from similar albeit reduced problems in delaying sister chromosome separation (in the *ter* region). Other ParB/*parS* systems should be tested in the same setup to make sure that the most appropriate system is used in future studies. In addition, sister chromatid interactions could be measured in the presence and absence of ParB-MT1 (and P1 ParB and FROS) using non-imaging tools (such as the site-specific recombination assay).

At the end of the discussion, the authors propose that ZapB binding to MatP/*matS* may exclude interaction with non-specific DNA. The authors have two assays available to measure non-specific DNA interaction by MatP/*matS*. To connect observation made *in vivo* and *in vitro*, the investigation of ZapB effects in the MatP DNA interaction studies would be a great addition to the manuscript and might simplify the interpretation of the *matP* mutant phenotypes.

The pull-down experiments shown in Figure S6 should be included in Figure 4 so that the reader is better able to discriminate non-specific from specific effects. Furthermore, a control reaction with 1 or 2 *matS* on DNAb and no *matS* on DNAf should be included as in the reverse case (in Figure S6) DNAf will compete with DNAb for non-specific binding. Can longer DNA molecules be included to exacerbate the non-specific binding effects?

Minor points:

The colour scheme for legends and graphs do not seem to match in Fig. 2c.

Fig. 2a: Drawing a bead bound to DNAb would help the reader to understand that DNAb is immobilized prior to interaction with MatP and DNAf.

We would like to thank all three reviewers for the time and experience in reviewing our manuscript. We have considered their comments and suggestions with great interest, taken them into account and included the necessary modifications in the main text of the manuscript. They have been highlighted in the manuscript resubmission for an easier reading.

Please find below our detailed, point-by-point responses, which we hope you will find satisfactory.

Best regards,

E. Crozat-Brendon on behalf of all authors.

Reviewer #1 (Remarks to the Author):

The authors combined *in vivo* and *in vitro* approaches to investigate the role of MatP in organizing the terminus region. *In vivo*, they used time-lapse microscopy to monitor the mobility of *ter* region. *In vitro*, they used DNA pull down assay and tethered particle motion assay to study how MatP bridges DNA. Their major conclusions are: the terminus region is not less mobile per se, but only less mobile when replicated *ter* foci are paired; After testing $\Delta matP$, $matP\Delta 20$ and $\Delta zapB$, they found that MatP's tetramerization domain, and MatP's interaction with ZapB are important for post-replicative pairing of *ter* foci. Another (unknown) activity of MatP is also required for post-replicative pairing of *ter*, and the authors proposed that this other activity could be related to MatP excluding MukB from the terminus region, which may lead to lower TopoIV at *ter* to remove precatenanes (Nolivos et al NatComm 2016). *In vitro* assays did not find MatP specifically bridging different *matS* sites, but found that MatP bridges *MatS* and non-specific DNA.

MatP is an important protein for chromosome organization and is relatively well studied. Previous work from several labs found that 1) MatP reduces the mobility of the terminus region, and was proposed to do so through bridging different *MatS* sites; 2) MatP interacts with the ZapB protein and 3) MatP excludes MukB from the terminus region. This manuscript extends previous findings and clarifies that the reduced *ter* mobility is only seen in paired foci, and proposed that MatP does not act to bridge different *MatS* sites but bridges between *matS* site and the ZapB protein. These hypotheses provide new insights into the mechanism of MatP action, but direct experimental evidence for some of these claims are missing (see below).

Major concerns:

1) A novel hypothesis for this paper is that, a tetramer of MatP does not bridge different *matS* sites, but bridge between *matS* and ZapB. Direct experimental evidence for this hypothesis is missing. There might be some experiments the authors can perform to test this. For *in vitro* experiment, perhaps the authors can use a similar experiment as in Figure 4, to use MatP bound to *matS* to pull down ZapB protein instead DNA.

We agree with the referee on this point. We tried different experiments to have direct evidence of an interaction between *matS*, MatP and ZapB. First, we tried a "supershift assay". This assay is based on the EMSA presented in Figure 6 but in presence of purified ZapB. Results show no difference in presence or absence of ZapB, suggesting that ZapB does not interact with *matS*/MatP complexes. We have included this result in the manuscript (Supplementary Fig. 10). This negative result is maybe

not fully convincing since EMSA only characterize stable complexes. We thus performed, as proposed by reviewer1, a pull-down experiment comparable to the one presented in figure 4 but in presence of ZapB. For this experiment we used two DNA molecules (DNA_b & DNA_r), each containing two *matS* sites. If pairing of the two DNA molecule by MatP is observed in this new experiment (Figure S11 a and b), no ZapB is pulled down (Figure S11 a and b, western blot or SDS-PAGE analysis of the pull-down). These results also suggest that there is no detectable interaction between *matS*/MatP and ZapB. Although these negative *in vitro* results do not rule out the fact that *matS*/MatP/ZapB interactions may occur *in vivo*, we modified the discussion of the paper in their light.

For *in vivo* experiment, maybe the authors could perform a ChIP-seq experiment, using antibodies against ZapB, to see whether ZapB is specifically enriched at *matS* sites, and whether the enrichment is dependent on the tetramerization domain of MatP.

We agree that this would be an interesting experiment to do. However, the complex FtsZ/ZapA/ZapB/MatP/*ter* has been extensively studied *in vivo* (Espeli et al., 2012; Buss et al., 2013, 2015, 2017; Männik et al. 2016). It clearly came out from these studies that the *ter* domain (*matS* containing domain of the chromosome) colocalizes with ZapA/ZapB structure in a MatP dependent manner. We assume that these data are enough to argue that a ZapA/ZapB/MatP/*matS* form a meta-complex *in vivo*. The open question is on the structuration of this complex and why it is not visible *in vitro*. We have included this point in the discussion.

2) The authors proposed that the remaining pairing activity of *ter* in MatP Δ 20 involves the control of sister chromosome decatenation, that is MatP Δ 20 could exclude MukB from *ter* leading to lower TopoIV levels to decatenate the replicated *ter* (Nolivos et al NatComm 2016). This hypothesis could be tested experimentally. First, the authors could mildly express TopoIV (See Wang et al GeneDev 2008; Lesterlin et al EMBOJ 2012) in WT or in MatP Δ 20 to test whether *ter* pairing is reduced. Second, the authors could inactivate TopoIV by using *parCts* or *parEts* (See Wang et al GeneDev 2008; Lesterlin et al EMBOJ 2012) to test whether other loci (like *ori*) now exhibit a pairing phenomenon similar to the *ter* region.

This is a very good suggestion, and we have indeed expressed TopoIV from the same plasmid pWX35 as Wang et al 2008. This shows indeed that the difference in FH number between our Δ *matP* strain and our MatP Δ 20 and Δ *zapB* strains is indeed due to decatenation. These results have been added in the results part (Fig 4) and the discussion.

However, despite multiple attempts, we were unable to transduce the strains with a *parETs* allele. As the results from expressing TopoIV are nicely clear, we do not consider this second experiment as essential and hope the referee will agree with us.

3) The authors used *parS*-ParB system to visualize the *ter* region. The cells would have 1 copy of *parS* before replication and two copies of *parS* sites after replication. We would expect to see foci with roughly 1x or 2x intensity. It is thus surprising to see that the distribution of fluorescent intensity of foci is rather smooth and not bimodal (figure 1a). Could the authors comment on this?

Deciphering when/if foci have been replicated has been a challenge since the beginning of the use of fluorescent labelling of foci (FROS and ParB-*parS* techniques). For technical reasons, nobody has ever been able to show a doubling in fluorescence for replicated foci. In our case, this is still true, and we think it is due to two major factors. First, the structure of the ParB-*parS* focus is a cage where ParB interacts with DNA and other ParB in a 3D manner (Sanchez 2015, Debaugny 2018). This does not necessarily imply a doubling in the quantity of ParB surrounding two *parS*. Second, in our

experiments, ParB is induced by a small quantity of IPTG, which results in a range of number of ParB per cell. We were therefore not expecting a bimodal distribution of intensities.

4) Previous studies (Bates and Kleckner, Cell 2005; Wang et al, GenesDev 2008; Joshi et al PLOSGen 2013) suggest that the origin region undergoes significant cohesion (between 10-15% of the cell cycle) after replication. We would expect to see 10-15% of the cells with 2x fluorescent intensity at origin. But the authors saw only 2% of the cells with high intensity. Could the authors comment on absence of origin cohesion in this study? In fact, the mobility of the cohesed origin foci would have been a nice control or comparison to the mobility of the paired ter foci.

We agree that observing cohesive origins would have been a nice control indeed. For some reason, the intensity of the Ori2 foci is lower than Ter4, so it is probable that origins are cohesive but the threshold of intensity to consider in this case is lower. As there is no drop of MSD in the MSD vs Intensity graph, it is very difficult to decide which threshold to consider. As explained earlier, we do not necessarily expect a doubling of intensity over replicated foci, making the decision rather arbitrary. Therefore, there are probably a non-negligible amount of cohesive origins in the 1FL group for Ori2, but we have unfortunately no way of knowing which ones they are.

5) In Fig S1, when using the pMT1 parS and P1 parS systems to label the origin, the fluorescence intensity distribution is the same for origin. However, the distribution of intensity for terminus changed dramatically. It seems that the P1 parS system is more invasive for the ter region and not invasive for the origin region. Could the authors comment on this discrepancy?

ParB-P1 has been seen as very invasive by other authors (Nielsen, 2006, ref#30; Espeli O, personal communication) and in our lab (Stouf et al, 2013). As stated earlier, these proteins (ParB-P1 and pMT1) form a nucleation point, or a cage, which is probably dependent on the DNA conformation and compaction around the *parS* site. For both systems, it looked like the foci were brighter at Ter than Ori, which could be explained by a greater affinity of ParB for the DNA as it is in Ter. At present, what makes the difference – e.g., differences in DNA concentration, catenanes structure, supercoiling at the post-replicative state, NAP binding (e.g., HU appears to have a special role in ter; Lioy et al., 2017) – remains unknown. Anyway, this has been observed in different labs.

Reviewer #2 (Remarks to the Author):

Comments on Crozat et al.,

This manuscript investigates the effects of MatP on the constraints on the chromosomal ter in E. coli. The conclusions are (i) ter loci are not intrinsically less mobile than ori loci, even in presence of MatP; (ii) however, a significant proportion of ter foci show a more intense fluorescence, most preferentially when they are located at mid-cell and are much less mobile; (iii) the proportion of these foci and their low mobility depend on all described activities of MatP; (iv) MatP binds matS-containing DNA as a tetramer but this complex contains only one DNA fragment; (v) bridging matS-containing DNA can be observed in vitro but is efficiently competed by non-specific DNA. (copied from the discussion).

The experiments described are very carefully executed and critically analyzed, which results in a number of interesting observations that support the conclusions summarized above.

Major comments

Differences were found between the P1 and the MT1 system. The Mt1 system was chosen for further experimenting. However, why this was chosen is not very clear. P1 has more cells with one ter focus compared to P2. What was the expected number of ter foci given the mass doubling time and the DNA replication and segregation rates? The position of the P1 ter locus in the chromosome is different from the MT1 ter focus, how are you sure that they behave similarly are these foci not interfering with the binding of other proteins and the segregation of the ters? Can you provide better argumentation why the MT1 system is better?

From published data of the same strains grown in similar conditions (M9 broth supplemented with glucose), we expected less than 50% of single Ori foci and about 80% of single Ter foci (e.g., see Mercier et al, 2008). This is what we obtained.

In our hands (and also our colleagues', though it is not well documented), the pMT1 system was the best to use because it generated less post-replicative cohesion at Ter. This had been observed and studied previously with more systems (*tetO/TetR*, Stouf et al 2013; Nielsen 2006, ref#30) and the ParB-pMT1 system has always been the system showing the less disturbance from a wild type cell: no effect on the cell cycle and reduced artificial cohesion compared to ParB-P1. We are confident that the difference seen is not due to the location of *parS*, since the same difference has been observed for other Ter (*ydgJ*, Stouf 2013) and Left loci (Nielsen, 2006).

Next the position, intensity and dynamics of the foci are analyzed from which it is concluded that highly intense foci are sister ter loci that are cohesive and constrained at midcell. The experiments were repeated in a cell deleted for MatP that forms the link between the septum and the ter regions by binding on the one hand ZapB and on the other *matS* on the chromosome. MatP appeared to be responsible for the confinement of the sister ters at mid cell as the number of FH decreases and the number of cells with 2 FL increases. The FH were more mobile in the absence of MatP but not in the absence of ZapB or in a *Matp20* that is not able to bind ZapB or form tetramers. Interestingly in the cells with a single and double ter FL, these foci (but not the ori FL) were more dynamic and less confined in the delta ZapB than in the delta MatP strain.

In the Discussion: Since ZapB self-assembles into large structures and clusters around ter (in a MatP-dependent manner) and the divisome (in a MatP-independent manner) 36, we suspect these clusters limit the diffusion of ter loci both in presence and absence of MatP. I agree with the observed data, but how logic is your explanation? ZapB is very negatively charged. How would ZapB constrain the ter region without interacting with MatP as DNA is also negatively charged? Are we may be missing a factor?

We agree with this comment, and have added the possibility at the end of the sentence: "we suspect these clusters limit the diffusion of *ter* loci via interaction with *matS*-bound MatP but also by a mechanism independent of MatP, yet to be described." (p13)

In the absence of MatP, about 5% single FH are left. If I remember it correctly about 12% of the chromosomes need to be deconcatenated by FtsK/Xer. Would the most simple conclusion not be that all single FH in delta MatP are concatenated chromosomes and that MatP is not involved in this part of the ter constrains?

We agree with this comment, and have added a sentence in the results to clarify this point: "However, a large fraction of remaining FH certainly contains paired foci. Indeed, sister Ter4 loci present on chromosome dimers may pair during FtsK/Xer processing resolving the dimer, even in absence of MatP, since chromosome dimer resolution does not depend on MatP⁹". (p7)

Next in vitro experiments using DNA with 0, 1 or 2 matS sites and isolated MatP. MatP is binding in general to DNA, but with higher affinity when matS is present. It is able to bind two strands of DNA. MatP is able to loop DNA that has two matS sites. IN a TMP set up the tetrameric MatP binding to a DNA strand causes it to shorten i.e change its conformation, whereas the dimeric MatP20 has no effect on the DNA conformation. MatS is not required for these changes. MatS sites are required for looping and not sensitive to competition by matS-less DNA. But pairing of distant DNA loci by MatP is sensitive to non-specific DNA competition. Therefore, pairing of matS sites might not be relevant in vivo.

Minor comments

Introduction :

Page 4. MatP-bound sister ter . Should it not be bound?

Results:

Page 5: The MSDs obtained with the pMT1 system are higher (larger?) than with the P1 system

Page 5: unsegregated (single) foci being closer from (meaning?) mid-cell than segregated (double) ones.

Page 5: Single foci of Ter4 are located closest (closer?) to mid-cell than double foci

Figure 2C the green line is of a different color as the legend green square.

Page 6: The effective diffusion coefficient is higher for Ori2 than Ter4 (3.5×10^3 and $2.9 \times 10^3 \mu\text{m}^2/\text{s}$, respectively), showing that despite a higher condensation, Ori2 is freer to diffuse than Ter4. Can you give the significance of the difference?

Page 6: Compared to the values obtained for the mat locus in budding yeast ($90 \text{ kBT}/\mu\text{m}^2$, 32). Can you give a bit of information on the mat locus so that the reader understand why you are comparing these two loci.

All these comments/corrections have been done in the text.

Page 7: This difference might be explained by the higher intensity of FH, which should reduce their mobility 35. Assuming that a FH is 70 molecules of GFP and a FL is 33 GFP using the information of the cited paper, you can probably calculate whether the expected differences are the same as the data you observe. Is the difference between FL and FH indeed due to the increase in the number of FP or could the interaction with FtsK play a role in a subset of the FH foci?

This sentence has been modified to point towards a role of FtsK/XerCD (Same as major comment #3: "Indeed, sister Ter4 loci present on chromosome dimers may pair during FtsK/Xer processing resolving the dimer, even in absence of MatP, since chromosome dimer resolution does not depend on MatP⁹". (p7)).

It is difficult to extrapolate information from this other paper as they did not calibrate the number of GFP/focus, it is not the same ParB system (theirs is P1, ours is pMT1) and not exactly the same GFP – ours is yGFP.

Page 8: This suggests that the interaction of MatP with ZapB is the main reason for FH foci formation, although the tetramerisation of MatP and a third activity that certainly involves an interaction with MukB are probably also involved. How can you have a certain third activity that probably is involved?

We have changed this sentence, which was indeed unclear.

We inferred that a third activity is involved, because the MatPD20 mutation cancels the activity of tetramerisation and interaction with ZapB. Nevertheless, the results obtained with the *matPD20* strain are not comparable to the ΔmatP strain. The difference left is probably due to a third activity

of MatP, which is probably related to its interaction with MukBEF/TopoIV (Nolivos 2016). We now show that a slight overexpression of TopoIV in the MatP Δ 20 as well as in the Δ zapB strains lowers the level of FH to the same as in Δ matP strain. We can then now be sure that we have identified all the activities required for MatP to hold the FH foci. The results and discussion have been modified in the light on these results.

Figure 4b. In this figure DNAb and DNAf is observed. Apparently, also the DNA from the magnetic bead is eluted. Why is this much less the case in Fig 4c? Should the elution of the bead not be independent of MatP?

A slight elution of DNAb was indeed observed in some experiments; this varied among experiments, but it is independent of MatP: the quantity eluted remains essentially the same for all concentrations of MatP on a same gel (see Fig 4b, and S6)

Reviewer #3 (Remarks to the Author):

The manuscript by Crozat et al. investigates the role of MatP protein in the spatial organization and temporal dynamics of the E. coli chromosome. The work shows by live cell imaging that a selected locus in the terminus region exhibits two distinct behaviours: normal mobility (similar to a control locus) and constrained mobility. The authors suggest that the constrained mobility is caused by local sister chromosome cohesion which is in large parts due to MatP protein and matS sites. DNA-DNA bridging by MatP, MatP/matS binding to ZapB and exclusion of MukBEF activity from the terminus region contribute differently to constraining the mobility of the terminus region. Finally, the authors provide biochemical experiments with purified MatP protein and DNA molecules to characterize the DNA bridging activity of MatP.

The work is executed very carefully and presented in a precise and easily accessible manner. The main conclusions of the manuscript are well supported by the experimental data, and potential pitfalls of the observations are openly discussed. Altogether, the manuscript provides a very solid piece of work with new technical and biological insights. Few points detailed below should be considered prior to publication.

Main comments:

The characterization of different visualization systems provides valuable information for the design of future imaging experiments in different model organisms. While ParB from pMT1 seems to report the behaviour of chromosomal loci more faithfully than the widely used FROS or P1 ParB systems, it may nevertheless suffer from similar albeit reduced problems in delaying sister chromosome separation (in the ter region). Other ParB/parS systems should be tested in the same setup to make sure that the most appropriate system is used in future studies. In addition, sister chromatid interactions could be measured in the presence and absence of ParB-MT1 (and P1 ParB and FROS) using non-imaging tools (such as the site-specific recombination assay).

As replied to rev#1: "This had been observed and studied previously with more systems (*tetO/TetR*, Stouf et al 2013; Nielsen 2006) and the ParB-pMT1 system has always been the system showing the less disturbance from a wild type cell". We feel it would be very time-consuming and go slightly out of the purpose of this paper to identify and test a whole collection of labelling systems. This remains obviously a very important question that we will address in future studies.

At the end of the discussion, the authors propose that ZapB binding to MatP/matS may exclude interaction with non-specific DNA. The authors have two assays available to measure non-specific DNA interaction by MatP/matS. To connect observation made *in vivo* and *in vitro*, the investigation of ZapB effects in the MatP DNA interaction studies would be a great addition to the manuscript and might simplify the interpretation of the matP mutant phenotypes.

As replied to rev#1 (major comments 1 & 2) on this point, we have done new experiments to study the effect of ZapB on MatP/matS interactions (Sup Fig. 10 & 11). None of these *in vitro* experiments shows any impact of the presence of ZapB on MatP/DNA interaction. However, this does not rule out the fact that *matS*/MatP/ZapB interactions occurs *in vivo* has proven by many studies (Espeli et al., 2012; Buss et al., 2013,2015,2017; Männik et al., 2016). Why it is not visible *in vitro* remains an open question for future projects. We have included this point in the discussion.

The pull-down experiments shown in Figure S6 should be included in Figure 4 so that the reader is better able to discriminate non-specific from specific effects. Furthermore, a control reaction with 1 or 2 matS on DNAb and no matS on DNAf should be included as in the reverse case (in Figure S6) DNAf will compete with DNAb for non-specific binding. Can longer DNA molecules be included to exacerbate the non-specific binding effects?

Figure S6 has been included in Fig 4 and the control reaction in Fig S6. The DNA molecules are already very long (1700 and 3800bp) compared to a *matS* (23bp), and 100nM of competitors are added to all reactions. According to our calculations, we therefore have an 44x excess of 23bp, non-specific site for each *matS* in the case of 2 *matS*/DNAb and DNAf, in the form of long and short DNA. We think this is enough to compete efficiently with specific binding.

Minor points:

The colour scheme for legends and graphs do not seem to match in Fig. 2c.

This has been modified.

Fig. 2a: Drawing a bead bound to DNAb would help the reader to understand that DNAb is immobilized prior to interaction with MatP and DNAf.

This was a mistake that has now been corrected.

REVIEWERS' COMMENTS:

Reviewer #1 (Remarks to the Author):

While I found that most of my concerns were addressed satisfactorily, my major concern about the ParB/parS system remains. In fact, all three reviewers raised concerns about the ParB/parS system used for visualization. The authors used foci of high fluorescence intensity (FH) as indication for "post-replicative pairing" of chromosome regions. However, as the authors stated in the rebuttal, the fluorescence intensity does not directly correlate to the replication status. Therefore FH could also mean that those cells happen to have a higher concentration of ParB-GFP, causing not only higher fluorescence intensity but also greater caging effect and slower mobility observed. This explanation is consistent with the observations that 1) the frequency of FH is lower at ori than at ter, likely due to more ori copies for ParB-GFP to distribute to; and 2) the lack of effect for TopoIV+ in WT (Figure 4): The authors proposed that post-replicative pairing is due to reduced decatenation by TopoIV at the terminus region resulting from MatP displacing MukB from ter. However, overexpression of TopoIV did not convert the FH to FL (figure 4, compare T4 and T4-TopoIV+), which does not support the authors hypothesis but support my earlier explanation of higher ParB-GFP concentration. This is a major issue for the interpretation of the results that lead to the central conclusion of this paper. The authors should add a couple of sentences to clarify that while FH is consistent with post-replicative pairing of foci, the authors cannot rule out the possibility that FH is due to higher concentration of ParB-GFP in some cells causing greater caging effect and reduced foci mobility.

Reviewer #2 (Remarks to the Author):

Comments on 222557

A MatP tetramer binds only one matS site in the in vitro experiments (unless two are on one strand of DNA). The authors assume that MatP is either binding other parts of the DNA non-specifically or ZapB. Adding ZapB does not provide any evidence for binding MatP-matS in vitro. ZapB is very negatively charged and so is the DNA, by using a long strand of DNA with a single matS site, a considerable repulsion is created to avoid ZapB binding. Did the authors try adding counter ions? (yes, they did 1 mM MgCl₂). If that is not enough, would it be possible to use a short strand of DNA with a matS site. It may not be an enormous amount of extra work if some MatP and ZapB are still in the fridge and it would be very satisfying being able to reproduce the in vivo evidence in vitro. DNA in the cells is probably covered with many proteins making perhaps it charge less repulsive for ZapB? In addition, the Z-ring through ZapA is concentrating ZapB at midcell, which might add to its ability to bind MatP despite its negative charge.

The same argument could be for the two DNA molecules with different fluorophores, they could also repulse each other, making it very difficult for MatP to bind both strands (with only one containing a matS site).

Line 843 page 25: This experiment is performed like other pull-down experiments but with different MatP concentrations with or without ZapB . How much ZapB?

Reviewer #3 (Remarks to the Author):

The authors have improved the manuscript during the revision, mostly by making minor modifications to the text and by adding control experiments. The overexpression of topo IV is helpful. It clearly supports the notion of multiple independent pathways for chromosome pairing. Following minor points should be considered prior to publication.

Minor points:

Line 271: 'is not mediated by catenane persistence'

Line 422: 'depends on ZapB but not on catenanes'

These conclusions are not strictly valid as catenanes may persist despite topo IV overexpression in wt MatP cells (the elimination of putative catenanes is not demonstrated here). Please re-word.

'was not affected by increased topo IV levels' or similar

Line 457: typo: 'in vitro' should be 'in vivo'.

Same line: 'the complex...is extensively documented in vivo' seems inappropriate/imprecise as physical interactions mediating complex formation cannot be inferred from localization/imaging data. Please reword. 'multiple findings point to the existence of ...' or similar.

The title could be more declarative by replacing 'sister chromosomes' for 'sister terminus regions' (or similar).

Dear reviewers,

We would like to thank you all again for your time in reviewing and improving our manuscript. We have considered your comments and modified the main text accordingly. Please find below our detailed, point-by-point responses, which we hope you will find satisfactory.

Best regards,

E. Crozat-Brendon, on behalf of all authors.

REVIEWERS' COMMENTS:

Reviewer #1 (Remarks to the Author):

While I found that most of my concerns were addressed satisfactorily, my major concern about the ParB/parS system remains. In fact, all three reviewers raised concerns about the ParB/parS system used for visualization. The authors used foci of high fluorescence intensity (FH) as indication for “post-replicative pairing” of chromosome regions. However, as the authors stated in the rebuttal, the fluorescence intensity does not directly correlate to the replication status. Therefore FH could also mean that those cells happen to have a higher concentration of ParB-GFP, causing not only higher fluorescence intensity but also greater caging effect and slower mobility observed. This explanation is consistent with the observations that 1) the frequency of FH is lower at ori than at ter, likely due to more ori copies for ParB-GFP to distribute to; and 2) the lack of effect for TopoIV⁺ in WT (Figure 4): The authors proposed that post-replicative pairing is due to reduced decatenation by TopoIV at the terminus region resulting from MatP displacing MukB from ter. However, overexpression of TopoIV did not convert the FH to FL (figure 4, compare T4 and T4-TopoIV⁺), which does not support the authors hypothesis but support my earlier explanation of higher ParB-GFP concentration. This is a major issue for the interpretation of the results that lead to the central conclusion of this paper. **The authors should add a couple of sentences to clarify that while FH is consistent with post-replicative pairing of foci, the authors cannot rule out the possibility that FH is due to higher concentration of ParB-GFP in some cells causing greater caging effect and reduced foci mobility.**

We agree with the reviewer that a cell to cell variability of ParB production is certainly responsible for a part of the variability in fluorescence intensity and in foci mobility observed throughout our study. This is best illustrated by the monotonous decrease in mobility with increasing intensity we measured (also according to previous studies). However, using the pMT1-derived system, this relationship clearly changes for foci above 1000 AU (fig1f), from which we decided to define FH from FL and analyse them separately. This is consistent with an additional effect acting on mobility above 1000 AU in addition to the simple increase in intensity.

Obviously, both FH and FL follow gaussian-like distributions concerning their intensity, so that some foci counted as FH are just the most intense foci containing unpaired loci and

reciprocally. This is what we stated in the results: “Remaining FH may be rare single loci with high fluorescence intensity, which should reduce their mobility³⁶.”

While we agree that the lack of effect of TopoIV overproduction on Ter-borne foci may be surprising, we disagree with the interpretation proposed by the reviewer. Indeed, TopoIV overproduction has an important effect in *matP*Δ20 and Δ(*zapB*) strains (fig4). One would thus have to assume that these two mutations have the same effect on ParB production, which is most unlikely. Neither protein was reported to affect gene expression.

Lastly, we detected more cells with two Ter-borne FH than cells with one Ori-borne FH (e.g., fig 2b), strongly arguing against the hypothesis of ParB titration, by increased copies of the Ori compared to the Ter locus, explaining the faint number of Ori-borne FH.

A strong argument in favour of our pairing hypothesis is that FH are converted into FL in two conditions: (i) when MatP is deleted; (ii) when TopoIV is overproduced providing that the Ter-linkage activity of MatP is inactivated.

Reviewer #2 (Remarks to the Author):

Comments on 222557

A MatP tetramer binds only one *matS* site in the in vitro experiments (unless two are on one strand of DNA). The authors assume that MatP is either binding other parts of the DNA non-specifically or ZapB. Adding ZapB does not provide any evidence for binding MatP-*matS* in vitro.

ZapB is very negatively charged and so is the DNA, by using a long strand of DNA with a single *matS* site, a considerable repulsion is created to avoid ZapB binding. Did the authors try adding counter ions? (yes, they did 1 mM MgCl₂). If that is not enough, would it be possible to use a short strand of DNA with a *matS* site. It may not be an enormous amount of extra work if some MatP and ZapB are still in the fridge and it would be very satisfying being able to reproduce the in vivo evidence in vitro. DNA in the cells is probably covered with many proteins making perhaps its charge less repulsive for ZapB? In addition, the Z-ring through ZapA is concentrating ZapB at midcell, which might add to its ability to bind MatP despite its negative charge.

The same argument could be for the two DNA molecules with different fluorophores, they could also repulse each other, making it very difficult for MatP to bind both strands (with only one containing a *matS* site).

We have also been disappointed not to see any effect of ZapB on MatP-DNA binding.

We do not think that the length of DNA matters that much. Indeed, the *matS*-containing DNA used in EMSA is very short (41 bp) and we failed to detect any effect of ZapB. In addition, DNA repulsion is certainly not strong enough to counter its pairing by bound proteins. We were able to detect both specific and non-specific bridging with long DNA (about 1700 and 4000bp in pull-down experiments and 2300 bp in TPM). We are thus confident we would have detected ZapB binding with the MatP-bound fluorophore-tagged DNA (41bp) if stable enough.

We suspect that formation of a DNA-MatP-ZapB complex might be a complex case. It may require the presence of ZapA and/or FtsZ to reproduce a filament that might be able to capture MatP-*matS*. Indeed, the (weak) MatP-ZapB interaction detected in vivo using the bacterial two hybrid system (Espeli et al., 2012) depended on ZapA.

Line 843 page 25: This experiment is performed like other pull-down experiments but with different MatP concentrations with or without ZapB . How much ZapB?

ZapB concentration ranged between 2 to 20uM. This has been added in the text.

Reviewer #3 (Remarks to the Author):

The authors have improved the manuscript during the revision, mostly by making minor modifications to the text and by adding control experiments. The overexpression of topo IV is helpful. It clearly supports the notion of multiple independent pathways for chromosome pairing. Following minor points should be considered prior to publication.

Minor points:

Line 271: 'is not mediated by catenane persistence'

Line 422: 'depends on ZapB but not on catenanes'

These conclusions are not strictly valid as catenanes may persist despite topo IV overexpression in wt MatP cells (the elimination of putative catenanes is not demonstrated here). Please re-word. 'was not affected by increased topo IV levels' or similar

We have replaced "catenanes" by the proposed sentence in the 1st case (line 27), and have added another sentence in the discussion (line 422: "assuming that over-expressing TopoIV leads to catenanes removal").

Line 457: typo: 'in vitro' should be 'in vivo'.

We thank the reviewer for spotting this mistake.

Same line: 'the complex...is extensively documented in vivo' seems inappropriate/imprecise as physical interactions mediating complex formation cannot be inferred from localization/imaging data. Please reword. 'multiple findings point to the existence of' or similar.

This sentence has also been reworded as proposed.

The title could be more declarative by replacing 'sister chromosomes' for 'sister terminus regions' (or similar).

We thank the reviewer for the helpful comments. All these points have been changed in the manuscript.